# Improved In Vivo Anti-Tumor and Anti-Metastatic Effect of GnRH-III-Daunorubicin Analogs on Colorectal and Breast Carcinoma Bearing Mice

**DOI:** 10.3390/ijms20194763

**Published:** 2019-09-25

**Authors:** Ivan Ranđelović, Sabine Schuster, Bence Kapuvári, Gianluca Fossati, Christian Steinkühler, Gábor Mező, József Tóvári

**Affiliations:** 1Department of Experimental Pharmacology, National Institute of Oncology, 1122 Budapest, Hungary; ivan.randel@gmail.com; 2Faculty of Science, Institute of Chemistry, Eötvös Loránd University, 1117 Budapest, Hungary; sabine.schuster83@gmail.com (S.S.); gmezo@caesar.elte.hu (G.M.); 3MTA-ELTE Research Group of Peptide Chemistry, Hungarian Academy of Sciences, Eötvös Loránd University, 1117 Budapest, Hungary; 4Department of Biochemistry, National Institute of Oncology, 1122 Budapest, Hungary; zeuglodon@freemail.hu; 5Preclinical R&D, Italfarmaco SpA, 20092 Cinisello Balsamo (Milan), Italy; g.fossati@italfarmaco.com (G.F.); c.steinkuhler@italfarmaco.com (C.S.)

**Keywords:** targeted cancer therapy, drug delivery system, daunorubicin, gonadotropin releasing hormone III derivates, peptide drug conjugates, oxime linkage, in vitro and in vivo anti-tumor activity, anti-metastatic activity, cellular uptake, gonadotropin releasing hormone receptor expression

## Abstract

Among various homing devices, gonadotropin-releasing hormone-III (GnRH-III) peptide represents a suitable targeting moiety for drug delivery systems. The anti-tumor activity of the previously developed GnRH-III-[^4^Lys(Bu),^8^Lys(Dau=Aoa)] conjugate and the novel synthesized GnRH-III-[^2^ΔHis,^3^d-Tic,^4^Lys(Bu),^8^Lys(Dau=Aoa)] conjugate, containing the anti-cancer drug daunorubicin, were evaluated. Here, we demonstrate that both GnRH-III-Dau conjugates possess an efficient growth inhibitory effect on more than 20 cancer cell lines, whereby the biological activity is strongly connected to the expression of gonadotropin-releasing hormone receptors (GnRH-R). The novel conjugate showed a higher in vitro anti-proliferative activity and a higher uptake capacity. Moreover, the treatment with GnRH-III-Dau conjugates cause a significant in vivo tumor growth and metastases inhibitory effect in three different orthotopic models, including 4T1 mice and MDA-MB-231 human breast carcinoma, as well as HT-29 human colorectal cancer bearing BALB/s and SCID mice, while toxic side-effects were substantially reduced in comparison to the treatment with the free drug. These findings illustrate that our novel lead compound is a highly promising candidate for targeted tumor therapy in both colon cancer and metastatic breast cancer.

## 1. Introduction

Colorectal cancer (CRC) is the third most commonly occurring cancer in men and the second in women, with 1.8 million new cases diagnosed (10.6% of all cancers) and 0.9 million deaths in 2018, while histologically and genetically heterogeneous breast cancer (BC) is the most prevalent in women population, with over two million new cases in 2018 (25.4% of cancer in women) worldwide [1]. Most of these cancers are highly malignant and they frequently metastasize and spread to distant organs, such as lung, liver, bone and brain which is mainly responsible for their incurability [2,3]. Some therapeutic approaches, such as surgery or radiation therapy, are only efficient in the case of an early stage diagnosis, otherwise the response to the treatment is not sufficient or cause only a temporary relieve [4]. Therefore, chemotherapy is still one of the main tools for cancer treatment. However, the application of anti-cancer drugs has several drawbacks and it often leads to a deficiency of chemotherapy. Hereby, the main limiting factor is the lack of selectivity to tumor cells and the associated impact on healthy tissue that generally results in toxic side effects [5]. Apart from that, cancer cells are highly vulnerable to mutations, which facilitates the development of resistances towards anti-cancer drugs [6]. Targeted therapy represents a promising tool for the selective and efficient delivery of anti-cancer drugs to cancer cells to overcome the disadvantages of classical chemotherapy [7]. Since this approach provides valuable benefits, such as the ability to differentiate between healthy and affected cells, a more precise identification and attack on cancer cells and the reduction of side effects on normal cells, this cancer treatment regime is gaining in interest and importance worldwide [8]. Targeted tumor therapy is based on the discovery that different cell surface proteins and receptors are overexpressed on various cancer cells compared to normal tissues [9]. Due to this, native, as well as artificial ligands of these receptors can be used as targeting moieties to deliver anti-cancer drug selectively to tumor cells without affecting healthy tissues, which prevents systemic toxicity [10]. In the past decades, different peptide-based drug delivery systems (DDS) have been developed and investigated, representing an attractive strategy for targeted tumor therapy due to their advantages, such as small molecule size, high tissue penetration, low immunogenicity, favorable pharmacokinetic properties and economical synthesis [11,12].

These receptors can serve as targets for peptide-based drug conjugates due to the fact that receptors for the human hormone peptide gonadotropin-releasing hormone (GnRH) are not only present in pituitary, but also highly expressed on the cell surface of many different tumor types (e.g., breast, ovarian, endometrial, prostate, renal, brain, pancreatic and melanomas), while their presence on healthy tissues is limited [13,14]. Therefore, a wide range of GnRH-I agonists and antagonists with different modification of the primary structure were developed and their in vitro and in vivo anti-tumor effect was evaluated [15,16].

In addition to human GnRH-I (Glp-His-Trp-Ser-Tyr-Gly-Leu-Arg-Pro-Gly-NH_2;_ where Glp is pyroglutamic acid), the native analog GnRH-III (Glp-His-Trp-Ser-His-Asp-Trp-Lys-Pro-Gly-NH_2_) has gained in importance and it was intensively studied within the past years. This natural GnRH-I isoform was identified and isolated from sea lamprey (*Petromyzon marinus*) and characterized by Sower et al. [17]. It was shown that GnRH-III specifically binds to GnRH-Receptors (GnRH-Rs) on cancer cells, causing a direct anti-proliferative activity on many tumor types [18], while its endocrine activity is strongly reduced when compared to GnRH-I, both in vitro and in vivo [19,20]. Therefore, GnRH-III and its derivatives can be considered as selective and efficacious targeting moieties that specifically bind to GnRH-Rs on cancer cells and thus represent a promising approach especially for the treatment of hormone-independent tumors like colorectal cancer [21]. Due to that, different GnRH-III-based peptide carriers were developed and used as targeting moiety to deliver anti-cancer agents, like daunorubicin (Dau), selectively to GnRH-R expressing cancer cells [22,23].

In previous studies, a wide range of modifications on the primary GnRH-III sequence have been performed in order to increase the anti-tumor activity of the conjugates by eliciting a favorable influence on the GnRH-R binding affinity, stability, cellular uptake rates and drug releasing properties [24,25,26]. While taking into account that the side chain of ^8^Lys could be modified by conjugation without revealing the reduction of the receptor binding affinity and the anti-proliferative activity of GnRH-III, this lysine residue provides a valuable ligation site for cytotoxic payloads [27]. Moreover, it has been shown that anthracyclines like doxorubicin (Dox) or Dau, can be efficiently linked to the side chain of ^8^Lys by incorporation of an aminooxyacetic acid (Aoa) moiety and formation of an oxime bond [28]. This ensures a selective in vitro and in vivo anti-tumor activity, leading to an increased survival rate of treated animals, since toxic side effects caused by premature drug release are prevented [29,30,31]. Nevertheless, it has been shown that oxime-bond linked GnRH-Dau conjugates elicit a significant tumor growth inhibitory effect on colon carcinoma bearing mice [32].

Different approaches have been used to improve the anti-tumor activity of the conjugates and thereby achieve more potent analogues [33,34]. One of this approach was the replacement of ^4^Ser by ^4^Lys followed by acetylation of its ε-amino group which provides a more suitable structure for receptor binding, enhanced enzymatic stability, improved cellular uptake and increased in vitro and in vivo anti-tumor activity [35].

Further studies pointed out that the acylation of ^4^Lys with short-chain fatty acids lead to an additional improvement of the biological activity of the GnRH-Dau conjugates. In particular, the butyrylated derivative GnRH-III-[^4^Lys(Bu),^8^Lys(Dau=Aoa)] (**1**) (Figure 1) displayed not only in vitro, but also in vivo an increased anti-tumor activity [25,34].

Based on discoveries that various amino acid substitutions led to an increased anti-proliferative activity of GnRH-III derivatives on GnRH-R positive prostate cancer cell lines [36], similar, as well as related modifications have been applied and investigated for oxime bond containing GnRH-III-Dau conjugates, whereby an *N*-terminal sequence modification led to an improved anti-tumor activity [37]. In particular, the conjugate GnRH-III-[^2^ΔHis,^3^d-Tic,^4^Lys(Bu),^8^Lys(Dau=Aoa)] (**2**) (Figure 1), where ^2^His was deleted and ^3^Trp was exchanged by the unnatural amino acid d-Tic (d-1,2,3,4-tetrahydroisoquinoline-3-carboxylic acid), was selected as the best candidate, displaying an improved inhibitory effect on the growth of colon and breast cancer cell lines in vitro [37]. Further investigations revealed that the enhanced anti-tumor activity of conjugate **2** is mainly related to improved cellular uptake and accelerated delivery of the drug to its site of action. In addition, it could be shown that compound **2** is highly stable in human plasma, while the high binding affinity to GnRH-receptors on cancer tissue and the release of the active drug metabolite in lysosomes were not affected.

Due to these promising findings, the aim of the present study was to gain further information regarding the potential of our best compound **2** as GnRH-III based DDS for targeted tumor therapy. Therefore, detailed in vitro and in vivo studies on the inhibitory effect on the growth of cancer cells have been performed in direct comparison to our previous lead compound **1**.

In the present study, the in vitro anti-proliferative activity of **1** and **2** was investigated on 23 different cancer cell lines and normal human lung fibroblast. Based on these results, distinct cell lines were selected for cellular uptake studies of the compounds and to determine the GnRH-receptor expression level, whereby not only the mRNA level, but also the absolute, as well as the cell surface receptor level was analyzed. Next to the in vitro analysis, the in vivo anti-tumor and anti-metastatic activity of **1** and **2** was investigated in orthotopic breast and colon carcinoma-bearing mice to ensure high potency of our GnRH-III based DDSs.

## 2. Results

### 2.1. In Vitro Anti-Proliferative Activity of GnRH-III-Dau Conjugates and Free Dau

The anti-proliferative effect of the GnRH-III conjugates **1** and **2**, as well as free Dau, was tested on wide range of cancer cell lines from different origin and also on MRC-5 (human fibroblast) as non-cancerous control cell line. The data showed that both conjugates possess an anti-proliferative effect on all cell types (Table 1). Except for the ovarian cancer cell lines A2780 and OVCAR-8, conjugate **2** displayed higher anti-proliferative activity than conjugate **1**, depending on the type of cancer cells. The lowest activity was measured on PANC-1 pancreatic cancer cells, whereby a high IC_50_ value was also obtained on MRC-5 cells, showing selectivity of the conjugates for cancer cell lines. The obtained IC_50_ values of the conjugates vary mostly in the low micromolar range and were one to two order of magnitude higher when compared to free Dau that can enter cells non-specifically by passive diffusion. Moreover, the relative potency was calculated as a ratio of conjugate’s IC_50_ and free Dau’s IC_50_ in order to show the potency of the conjugates independently from the cell line, due to different activity of free Dau. A lower value of relative potency indicates that the conjugate’s IC_50_ value is closer to the free Dau’s IC_50_ value, which implies that the targeting capacity of the conjugate as well as its anti-tumor effect is stronger on a particular cell line, as compared to a cell line with higher relative potency. The BC cell lines showed good response to the conjugates by IC_50_ values, as well as by relative potency. Besides, the conjugates showed high anti-proliferative activity on mice CRC cell line C26, while the conjugates showed a moderate anti-proliferative activity on HT-29 human colon adenocarcinoma, but the relative potency was in the same range as for the BC cells.

### 2.2. mRNA Expression Level of GnRH-R

Based on the results of the in vitro anti-proliferative activity, we chose different cell lines (MDA-MB-231, HT-29, A2780, PANC-1, U87MG and MRC-5), where the conjugates **1** and **2** had either a high activity, or the lowest activity, to determine the GnRH-R expression level.

The GnRH-R mRNA expression of these cell lines was quantified via reverse transcription quantitative real time PCR (RT-qPCR). The amount of GnRH-R mRNA was higher in all cancer cell lines in comparison to the normal cell line MRC-5, except for U87MG cells, where it was slightly lower (Figure 2A, Appendix A). In MDA-MB-231, HT-29 and A2780 cancer cell lines, the level of GnRH-R mRNA expression was 7.3, 4.6 and 3.4 times higher in comparison to normal cell line, while only 1.8-fold higher expression was obtained for PANC-1 cell line.

### 2.3. Absolute Protein Expression Level of GnRH-R

The GnRH-R protein level of the same cell lines was quantified by western blot. The protein level of GnRH-R was higher in all cancer cell lines as compared to the normal cell line MRC-5, except for PANC-1 cells, where it was 0.3-fold lower (Figure 2B, Appendix A). The highest protein level was obtained for A2780, U87MG and 4T1 cells, with 3.8-, 3.6- and 3.5-fold higher protein levels than MRC-5 cells, while MDA-MB-231 and HT-29 cancer cell lines showed 1.2- and 1.7-fold higher level of GnRH-R protein expression.

### 2.4. Cell Surface Protein Expression Level of GnRH-R

The GnRH-R cell surface expression level of these cell lines was quantified via flow cytometry. The GnRH-R expression on the cell surface was higher for all cancer cell lines in comparison to MRC-5 cells (Figure 2C, Appendix A). The highest level of GnRH-R was obtained for MDA-MB-231 and A2780 cells, with 5.7- and 5.6-fold higher surface expression, followed by U87MG, 4T1, and PANC-1 cell lines with around four-fold higher expression than MRC-5 cells, while HT-29 cancer cell line showed two-fold higher level of GnRH-R expression.

### 2.5. Determination of Cellular Uptake of GnRH-III-Dau Conjugates

The cellular uptake of the conjugates was measured by flow cytometry on the tested cell lines. The obtained results displayed that the new conjugate **2** was taken up more efficiently than **1**, with 1.7–2.7 times higher uptake rates, depending on the cell line (Figure 2D, Appendix A). The normal cell line MRC-5, as well as PANC-1 cancer cell line showed two-fold lower uptake capacity in comparison to the other cancer cell lines.

### 2.6. Acute and Chronic Toxicity Studies of GnRH-III-Dau Conjugates

An acute toxicity experiment was performed for 14 days, whereby no significant change in body weight could be observed (Figure 3A) and also the general looking and behavior of experimental animals was adequate, even under dose of 50 mg/kg Dau content of conjugate **2**. Chronic toxicity experiments were also performed for 14 days and the animals were treated with both conjugates five times under a dose of 15 mg/kg Dau content and free Dau (1 mg/kg). Also, in this experiment, we could not observe a significant change in body weight (Figure 3B), general looking and behavior of the mice.

### 2.7. Effect of GnRH-III-Dau Conjugates and Free Dau in Orthotopic 4T1 Mice Breast Tumor Model In Vivo

The body weight in orthotopic 4T1 mice breast carcinoma bearing mice was partly changed during the treatment time. The animal body weights in the control and free Dau treated groups were slightly decreased, while the animals treated with conjugates showed an increase in body weight at the end of experiment, compared to the start (Figure 4A).

The anti-tumor effect of the GnRH-III conjugates **1**, **2** and free Dau was evaluated measuring the tumor volume in each group during the experiment. All treated groups showed significant inhibition of the tumor volume by approximately 19% as compared to the control group at the end of the experiment. (Figure 4B).

The effect of free Dau and the GnRH-III conjugates on the proliferation of primary tumors in 4T1 orthotopic model was evaluated counting the percentage of KI-67 (proliferation marker) positive cells in comparison to all cells per field of view (magnification 400×) and calculating the proliferation index (Figure 4C). It was observed that both GnRH-III conjugates **1** and **2** caused a significant decrease of the proliferation index by 16.3 and 25.9%, as compared to the control, while free Dau decreased the proliferation index also significantly by 19%.

The effect of the GnRH-III conjugates and free Dau on the liver toxicity was evaluated measuring the liver weight at the end of the experiment and calculating the liver weight/body weight ratio (Figure 4D). The average liver/body weight ratio of the mice in the group that was treated with free Dau was significantly decreased by 9.8% compared to the control group, as well as in comparison to the liver/body weight ratio of mice treated with the conjugates which showed no significant changes in liver/body weights ratio.

The number of macro-metastases in peripheral organs, such as spleen, lung, liver and kidneys, was counted, in order to determine the anti-metastatic effect of free Dau and the GnRH-III conjugates on aggressive 4T1 BC orthotopic model (Figure 5A). The number of macro-metastases in spleen was significantly decreased in all treated groups (Dau, **1** and **2**) by 64.3, 72.8 and 78.1%. In the lung, the number of macro-metastases was also significantly reduced for all treated groups by 55.4, 55.2 and 64.4%, respectively. The numbers of macro-metastases in the liver and kidneys were decreased under treatments, whereby a significant decrease could be only obtained for conjugate **2**.

The effect of the GnRH-III-Dau conjugates and free Dau on the amount of micro-metastases and their proliferation in the lung (Figure 5B,C) was also evaluated by counting the number of micro-metastases and proliferating marker KI-67 positive cells and the calculation of their ratio in comparison to all cells per field of view (magnification 100×). The obtained data revealed that free Dau and both conjugates (**1**, **2**) significantly inhibited the number of micro-metastases in the lung by 33.7, 43.8 and 49.4%, as compared to the control group. The proliferation index of lung metastases was significantly inhibited by 27.8, 37 and 39.1% in groups that were treated with free Dau, **1** and **2**.

### 2.8. Effect of GnRH-III-Dau Conjugates and Free Dau in Orthotopic MDA-MB-231 Human Breast Tumor Model In Vivo

The effect of the GnRH-III conjugates **1** and **2**, as well as Dau on the animal body weight was evaluated in orthotopic MDA-MB-231 human breast carcinoma bearing mice (Figure 6A). In all the groups, the body weight was decreased at the end of experiment in comparison to the beginning. The body weight of the mice in control group was decreased by 2.7%, while in the groups treated with **1** and **2**, it was decreased by 10.1 and 8.2%. In comparison, free Dau caused a significant decrease of mice body weight by 20% and considering that two animals of the control group were in bad condition, the experiment was terminated on day 45 after cells inoculation.

The anti-tumor effect of the GnRH-III conjugates **1**, **2** and Dau was evaluated by measuring the tumor volume in each group during the experiment. All treated groups showed a significant inhibition of the tumor volume in comparison to the control group at the end of the experiment (Figure 6B). The treatment with Dau was most effectively, whereby the tumor volume was significantly inhibited by 46.3%. Apart from that, a significant inhibition of the tumor volume was also obtained in groups which were treated with conjugate **1** (34.1%) and **2** (23.1%). Moreover, from day 38 after cells inoculation, the inhibitory effect of free Dau was significant in comparison to control group.

The anti-tumor effect of the GnRH-III conjugates and Dau was evaluated also by measuring the tumor weight in each group after termination (Figure 6C). Based on these tumor weights, we determined that free Dau, **1** and **2** inhibited tumor weight significantly by 40.1, 28.7 and 27.7% in the case of orthotopic human MDA-MB-231 breast tumor model.

The effect of the GnRH-III conjugates and Dau on the liver toxicity was evaluated by measuring the liver weights at the end of experiment and calculating the liver weight/body weight ratio (Figure 6D). The average liver/body weight ratio in the group treated with free Dau was significantly decreased by 16.8% as compared to the control group. Conjugates treated groups showed non-significant changes in liver/body weight ratio.

The anti-metastatic effect of Dau and the GnRH-III conjugates was evaluated by counting animals containing metastases close to the primary tumor at the end of experiment (Table 2.). It could be observed that all animals in the control group had metastases close to the primary tumor. In group treated with free Dau, four out of seven mice had metastases, while for the groups treated with **1** and **2** the best anti-metastatic effect could be obtained with three out of seven animals with metastases.

### 2.9. Effect of GnRH-III-Dau Conjugates and Free Dau in Orthotopic HT-29 Human Colon Tumor Model In Vivo

The animal body weight in orthotopic HT-29 human colon carcinoma bearing mice decreased in all groups at the end of experiment when compared to the start (Figure 7A). The mice in free Dau treated group exhibit significantly decreased body weight, whereby the experiment was terminated on day 23 after tumor transplantation (day 17 of treatment). On the same day, the decrease of the body weights in the control and conjugates (**1** and **2**) treated groups were non-significantly lower. The body weight of the mice in the control group was significantly decreased on day 30 after tumor transplantation which was the reason for experiment termination. At the end of the experiment, the body weight of the groups treated with GnRH-III conjugates **1** and **2** was also significantly reduced.

The anti-tumor effect of the GnRH-III conjugates and free Dau was evaluated by measuring the tumor weight in each group after the termination of the experiment (Figure 7B). The obtained data reveal that Dau, **1** and **2** significantly inhibited the tumor growth, whereby the tumor weights were reduced by 84.3, 80.8 and 87.1%, as compared to the control group.

The average liver/body weight ratio of the mice in the group treated with free Dau was significantly decreased by 29.4% compared to the control group, as well as in comparison to the liver/body weights ratio of the mice in both conjugates treated groups (Figure 7C), showing the high toxicity of free Dau. The groups which were treated with the conjugates **1** and **2** showed non-significant changes in liver/body weight ratio.

We summarized all results and the corresponding specificity of each model in Table 3 in order to provide a comprehensive overview.

## 3. Discussion

When considering that receptors for many regulatory peptides are overexpressed in various tumor cells, compared with their expression in normal tissues, peptide based drug tumor targeting is a promising therapeutic approach for cancer and enables the specific delivery of anti-cancer drugs to cancerous cells [9]. Among various homing devices, GnRH-III peptide represents a suitable targeting moiety and different GnRH-III conjugates have been developed and used as DDS [10]. Moreover, GnRH-Rs were not only detected in cancers related to the reproductive system, but also to unrelated ones, such as colon and pancreatic cancer [38,39,40].

Recently, a novel promising GnRH-III-based DDS with an improved in vitro activity was developed in our laboratories [37]. We analyzed its activity on a wide set of cancer cell lines and also on MRC-5 normal cell line in comparison to our previous lead conjugate **1** to further characterize the potency of this new conjugate (**2**) [34]. Furthermore, the in vivo activity of both compounds in tumor bearing mice was determined to gain a deeper insight of their potency.

The in vitro anti-proliferative effect measurements revealed that both conjugates elicited a substantial growth inhibitory effects with IC_50_ values in the low micromolar range, whereby conjugate **2** displayed mostly a higher cytostatic effect than **1**. These results are in line with our previously reported results and confirm the high potential of our novel lead compound **2** [37]. In this study, we obtained higher IC_50_ values in CRC and BC cell lines than in the previous one. This might be related to the difference in the assays which have been used for cell viability determination [41]. In the present study we used a MTT assay and in previous studies a resazurin based assay was used. Furthermore, we obtained beneficial relative potencies of compounds **1** and **2** on breast, colon, malignant glioma, melanoma and ovarian cancer cell lines. On the contrary, we obtained relatively high IC_50_ and relative potency values in case of MRC-5 cells, which suggested that the conjugates are highly tumor selective, except for PANC-1 cells, where the IC_50_ value of the free Dau was also substantially higher than for the other cancer cell lines. Higher values of relative potency indicate a loss of potency of the conjugates with respect to free Dau [42,43].

It is well known that pancreatic cancers reveal very often multidrug resistance (MDR) towards a variety of classical anti-cancer drugs [44]. Miller et al. has been determined that the MDR of PANC-1 cells is mainly related to the presence of the multidrug resistance-associated protein (MRP) [45]. Moreover, it has been shown that MRP mediates the ATP-dependent efflux of anthracyclines, like Dau and other anti-cancer agents [46]. Based on these findings, it can be assumed that the low activity of the free Dau is mainly related to the efflux of the drug from cytosol directly after passive diffusion.

The endocytic uptake of Dau via conjugates might overcome the MRP-derived MDR in PANC-1 cells, as shown by Zheng et al. [47]. Therefore, it can be assumed that the Dau resistance of PANC-1, can be reduced when the drug enters the cells by an endocytic route which could not be obtained with our conjugates. Thus, the low activity of the conjugates on PANC-1 cells might be related with a lower receptor level even if other reports verified GnRH-R expression on PANC-1 cells [40]. We determined the GnRH-R expression on a selection of cell lines, where **1** and **2** had a high activity and also on the two cell lines with the lowest activity, to better interpret our results. Thus, we determined the GnRH-R mRNA, absolute protein level and cell surface receptor level of MDA-MB-231, 4T1, HT-29 and A2780, as well as PANC-1 and MRC-5 cells.

The GnRH-R mRNA expression level was analyzed using RT-qPCR. All cancer cell lines showed a higher level of GnRH-R mRNA expression than the normal cell line which is in accordance with previous studies [38]. The detected GnRH-R mRNA level in PANC-1 cells was substantially reduced compared to MDA-MB-231, HT-29 and A2780 cancer cells which might be an explanation for the high IC_50_ values on PANC-1 cells. On the contrary, no correlation between the GnRH-R mRNA level and the IC_50_ value could be obtained in the case of U87MG cancer cells, which exhibit just a low mRNA level of GnRH-R expression, but low IC_50_ values. This observation might be explained by post-transcriptional processes that are involved in the final synthesis of the protein leading to different levels of its expression. It is well known that mRNA and protein expression do not always correlate, thus we determined also the absolute GnRH-R protein level [48].

Whole cell lysates of the appropriate cell lines were analyzed using western blot studies in order to determine the GnRH-R protein expression level. Human β-actin was used as reference housekeeping gene to normalize the GnRH-R expression. Each sample was compared to MRC-5 cells by relative quantification to enable a better comparison of the obtained results. All cancer cell lines showed higher protein level of GnRH-R in comparison to normal cells, except for PANC-1. Moreover, these results are in line with the determined IC_50_ with exception of MDA-MB-231. For the biological activity not only the absolute GnRH-R level is of high relevance, but also the location of the receptor on the cell surface [49].

Thus, the GnRH-R expression level on the cell surface was analyzed while using flow cytometry. All of the cancer cell lines showed higher GnRH-R receptor surface level in comparison to the normal cell line. The obtained results are in accordance with previously published data for breast [50], ovarian [51], colon [39], brain [52] and pancreatic cancer [40]. Moreover, these results correlate very well with the determined IC_50_ values. However, both conjugates showed low cytotoxic effect on PANC-1 cells, although we obtained a high surface expression level of GnRH-R. This effect could depend on a number of factors next to the binding to receptors, including the rate of internalization, efficiency of the drug release, interaction of drug with its targets and the suitability of the drug for a certain tumor [11,12]. We performed cellular uptake studies to further evaluate this assumption.

The cellular uptake of conjugates was analyzed while using flow cytometry. All of the cancer cell lines showed higher uptake rates for the conjugates in comparison to normal cells. These results are in line with the GnRH-R expression level and the IC_50_ values. In accordance with previous results, the obtained cellular uptake rates of the new lead compound **2** were for each cancer cell line significantly higher than that of **1**, but not for the normal cell line MRC-5, where we could not obtain a significant difference between the conjugates. This indicates that the uptake capacity on cancer cells is higher for conjugate **2** which is in line with our previous data [37]. Apart from that, the cellular uptake rates on PANC-1 cells were for both conjugates lower than for the other cancer cell lines and in the same range as the rates for MRC-5. These results are in line with the low anti-tumor effect of the conjugates on these two cell lines and the GnRH-R expression level.

Taking all of the results into account, it can be proposed that the biological activity of the conjugates strongly depend on the expression of the GnRH-Rs which ensures the selectivity of the GnRH-III based DDSs and on cellular uptake capacity of the compounds.

We decided to analyze the in vivo anti-tumor activity of the conjugates on tumor-bearing mice due to the promising in vitro results of **1** and **2**. At the outset of these studies, we determined the stability of the compounds in mice plasma revealing that both of the compounds were stable for at least 24 h (Appendix A). Although both conjugates have been already proven to be stable in human blood plasma [37], this experiment provides valuable information and avoids misinterpretation of preclinical results that might be caused by differences between the enzymatic activity of laboratory animals and humans [53].

The analysis of the toxicity of new compounds on healthy mice is essential for the drug development process [54]. Animal weight and behavioral changes are the critical characteristics in toxicity testing as animals should be protected from stress and pain [55]. Thus, an acute toxicity study experiment was performed [56], which revealed that neither the body weights of the mice were significantly changed after administration of the compound, nor the general looking and behavior of experimental animals, even at a dose of 50 mg/kg Dau content of conjugate **2**. We performed also a chronic toxicity experiment [57], where animals were treated with both conjugates five times with a dose of 15 mg/kg Dau content and two times with free Dau (1 mg/kg), to further evaluate the effect of the compounds on healthy mice. After 14 days, we could not observe any significant change in the body weights, general looking and behavior of mice. Based on these results, it can be concluded that both conjugates can be used at this concentration for the treatment of tumor bearing mice, since this dose was not toxic for the animals.

Unfortunately, most breast cancers are invasive and frequently metastasizes to distant organs, including lung and liver [58]. There is currently no effective therapy for metastatic breast cancer. Therefore, the development of therapeutic agents to suppress metastatic breast cancer is of great significance. To determine the efficacy of anti-tumor drugs, the choice of a suitable in vivo model that mimics the initiation and progression of breast cancer is highly important [59]. A number of murine models have become available during the last two decades and syngeneic breast cancer murine models, including the 4T1 murine mammary cancer models, originally isolated as subpopulation derived from a spontaneously arising mammary tumor in BALB/c mice [60], are widely used to study the mechanisms of tumor growth and metastasis [61]. Orthotopic breast cancer models mimic the tumor microenvironment in an adequate way and they provide more tumorigenic and metastatic cancer cell population [62]. Moreover, these models represent beneficial tools to investigate the anti-tumor and anti-metastatic effects of various drugs according to their aggressive growth and high invasive nature [63]. Therefore, we performed the orthotopic implantation of 4T1 cells by injection in the mammary fat pad, which leads to the formation of primary tumors and subsequent metastatic growth, establishing a fast and quantitative method [64] to determine the in vivo anti-tumor activity of our conjugates **1** and **2**, and Dau. Moreover, we analyzed the anti-metastatic effect of the conjugates and free Dau, since the majority of deaths of breast cancer patients are related to tumor cell metastasis.

The change of the animal body weight during the in vivo experiment is an important parameter that shows the condition of drug treated and non-treated tumor bearing animal [65]. In 4T1 tumor bearing mice, we obtained decreasing body weights for the control and Dau group, while we could observe an increase in animal body weight in both conjugate treated groups, indicating that the conjugates did not cause toxicity and side effects on the animals during the treatment.

The anti-tumor effect of the GnRH-III conjugates **1** and **2**, as well as the free Dau was evaluated by measuring the tumor volume in each group during the experiment. We obtained a significant tumor volume inhibition of a very aggressive orthotopic 4T1 mouse tumor in Dau, as well as conjugates treated groups compared to the control group, at the end of experiment on day 28 which verifies the significant anti-tumor effect of the conjugates against syngeneic orthotopic breast tumor.

Apart from that, the expression of the KI-67 protein is strictly associated with cell proliferation and tumor progression, therefore we determined the proliferation index of KI-67 positive cells [66]. Our result display, for both conjugates, as well as free Dau a significant lower proliferation index in primary tumor when compared to the control group, which supports their significant inhibitory effect on the tumor growth.

Furthermore, not only the elicited anti-tumor activity is of high relevance for the success of an anti-cancer drug, but also the selectivity to cancerous cells and reduction of side-effects. Dau is known to be rapidly and widely distributed in tissues, whereby the highest levels were found in the liver, spleen, kidneys, lungs and heart [67]. Since the liver is the vital organ in metabolism of Dau, production of toxic intermediates that may trigger liver injury and impair with the liver function can increase the risk of toxicity [68]. Analysis of organ weight in toxicology studies is an important factor for the identification of potentially harmful effects of drugs [69], thus the liver weight/body weight ratio analysis provides better understanding of drug toxicity [70]. The free Dau caused a significant decrease of liver/body weight ratio and spleen weight in comparison to control and conjugates treated groups, revealing that the treatment with Dau caused toxic side-effect in mice. From the obtained data, free Dau decreased the spleen weight significantly by 27.2% when compared to the control group (Appendix A) and this is in line with the obtained liver toxicity of free Dau measuring the liver weights. In comparison, non-significant liver/body weight ratio and spleen weight changes could be detected in GnRH-III conjugates treated groups proving evidences for their selectivity and non-toxicity to healthy tissue.

The number of metastases close to the primary tumor and on distant organs are often clinical picture in breast tumors [71]. Conjugates and the free Dau both exhibited a decreased number of macro-metastases in peripheral organs, especially in the lung and the spleen, while the new conjugate **2** revealed also significantly decreased metastases in the liver. We further evaluated the effect of GnRH-III-Dau conjugates and Dau on the number of micro-metastases and on the proliferation of metastases in the lung in order to confirm the significant decrease of the number of macro-metastases. The number of micro-metastases was significantly reduced for all treated groups in comparison to the control group, especially in conjugates treated groups. Both conjugates and free Dau showed a significant inhibition of the proliferation index in lung metastases, which allows for drawing the conclusion that both conjugates possess an anti-metastatic activity.

The significant in vivo tumor growth inhibition in 4T1 BC bearing mice, together with the significant inhibition of the proliferation in primary tumor and the significant anti-metastatic activity, supports the assumption that both conjugates are promising therapeutics for breast carcinoma without causing significant side-effects and toxicity.

Further studies have been performed in orthotopic MDA-MB-231 human BC bearing mice, because the use of human breast cancer cell lines in orthotopic xenograft mouse model provide a powerful tool for therapeutic investigation of anti-cancer agents and give valuable information about specific drug targeting in human [72]. In addition, this triple negative BC cell line is extremely aggressive, expresses a high level of GnRH-R and it has been shown to be a suitable model to evaluate the efficiency of GnRH peptide drug conjugates [73,74,75]. This prompted us to determine the in vivo anti-tumor activity of both GnRH-III compounds in orthotopic MDA-MB-231 human breast tumor bearing mice.

In this model, the animal body weight of the mice was decreased in all groups at the end of experiment when compared to the beginning. But, only in the group treated with free Dau, the body weight of the mice was significantly decreased. Taking into account that two animals in the control group were in bad condition, but not the mice in the groups treated with the conjugates, it can be concluded that the conjugates did not cause a substantial toxicity, even if the administrated Dau content was much higher than the maximum tolerated dose of the free drug [34].

If we consider the tumor volume inhibition, the strongest effect was obtained for the treatment with free Dau, but also both conjugates significantly inhibited the tumor volume. The anti-tumor effect of the GnRH-III conjugates and free Dau was evaluated also measuring the tumor weight in each group after termination of the experiment. Also, these data demonstrate that the conjugates **1** and **2**, as well as free Dau possess a significant anti-tumor activity in orthotopic human MDA-MB-231 breast cancer bearing mice.

In accordance with the results that were obtained in the 4T1 mouse breast model, the treatment with free Dau caused a significant decrease of the liver/body weight ratio compared to the liver/body weight ratios of the control and conjugates treated groups. When considering that we did not detect a significant change in liver/body weight ratio, it can be suggested that the treatment with the GnRH-III conjugates do not cause harmful side effects.

It has been reported that, in SCID mice, the remaining innate immune cells reduce the metastasis formation in distal organs [76]. Due to this, the anti-metastatic effect of the conjugates and free drug was evaluated based on the number of animals containing metastases close to the primary tumor. The obtained results provide evidence that both conjugates cause a significant reduction in the number of animals with metastases. Moreover, the anti-metastatic effect of the conjugates was higher in comparison with the Dau treated group, suggesting that these two conjugates are potential anti-metastatic therapeutics for BC.

Based on these results, we can conclude that both GnRH-III conjugates, as well as Dau inhibit the tumor growth significantly in MDA-MB-231 BC bearing mice. Furthermore, the anti-metastatic activity of the conjugates was significant and higher than that of Dau. Moreover, the results of the animal body weight, as well as the liver/body weight ratio, indicate that no toxicity side effects are caused by the treatment with the conjugates, even if the Dau content of the injected dose was much higher than the maximum tolerant of Dau.

Next to the in vivo studies on BC bearing mice, we were interested to analyze the anti-cancer activity of the two lead compounds on tumors that are not related to the reproductive system. Due to our in vitro results and previous in vivo studies of **1**, CRC might represent an adequate model for these studies [34]. Although orthotopic colon cancer xenograft models are technically challenging and labor-intensive, orthotopic transplants are able to more accurately mimic human tumors. This approach simulates better the natural microenvironment for tumor development, providing an effective approach to investigate tumor pathophysiology and to develop therapeutic strategies which allow a better prediction of patient’s response to chemotherapy in comparison with heterotopic transplants [77]. Apart from that, different studies pointed out that many colon tumors possess an increased expression level of GnRH-R [39], therefore a variety of synthetic therapeutics have been used which target these receptors and hence revealed significant tumor growth inhibition in vitro and in vivo [22,24,26,32,34]. Thus, HT-29 human colon tumors were implanted to the intestine of immunodeficient SCID mice [34,78].

We observed that the free Dau cause a significant decrease in mice body weights, which compelled us to terminate the experiment for this group already on day 23 after tumor transplantation. Apart from that, the significant loss of the body weight in the control group also prompted us to terminate the experiment on day 30 after tumor transplantation. A reduction of the animal weight was also obtained for mice which were treated with the GnRH-III conjugates, but here the effect was not that serious, especially on the day when the Dau group was terminated. This indicates that both conjugates cause less harmful side effects than Dau. Moreover, it might be possible that a distinct decrease of the body weight was caused by a higher susceptibility of the animals after surgery procedures, which was necessary for establishing the orthotopic colon cancer model [78].

The anti-tumor effect of the conjugates and Dau was evaluated after isolation of the tumors at the end of experiment. We obtained a significant inhibition of the tumor weight in all treated groups, whereby the highest inhibition was obtained for the treatment with our novel lead compound **2** even the experiment took longer in comparison with the free drug. Furthermore, this effect could be achieved without substantial toxic effects and by a lower dose than in our previous studies with **1** and free Dau [34].

In addition, we did not detect significant changes in the liver/body weight ratios of the groups treated with the conjugates, while a significant decrease of the liver/body weight ratios was observed for the group that was treated with the free drug. This indicates that the conjugates did not cause toxicity to the mice, unlike free Dau.

Based on these results, we can conclude that both GnRH-III conjugates, as well as free Dau inhibited the tumor growth efficiently in HT-29 CRC bearing mice. However, our new conjugate **2** showed the highest anti-tumor effect against CRC, while its impact on the animal body weight and liver/body weight ratio was the lowest demonstrating that compound **2** is a promising candidate for the targeted chemotherapy of CRC.

## 4. Materials and Methods

### 4.1. Synthesis of GnRH-III-Dau Bioconjugates

The GnRH-III-Dau conjugates **1** and **2** were prepared by a combination of solid phase peptide synthesis of the peptide carrier and oxime bond formation in solution, as previously described [37]. The compounds were purified by RP-HPLC using linear gradient elution with eluent A (0.1% trifluoroacetic acid (TFA)/water) and eluent B (0.1% TFA/acetonitrile (ACN)-water 80:20, *v*/*v*). The freeze-dried bioconjugates were used, without changing the TFA counter ions, for the in vitro and in vivo studies, in order to evaluate their tumor growth inhibitory effect. The free Dau (as HCl salt) and synthesized GnRH-III-Dau conjugates were dissolved in sterile water (Pharmamagist Kft., Budapest, Hungary) and used for the in vitro and in vivo studies.

### 4.2. Cell Lines and Culture Conditions

In experimental procedures following cell lines were used, MDA-MB-231 and MCF-7 (human breast cancer), 4T1 (mice breast cancer), DU145 and PC-3 (human prostate cancer), A2780, OVCAR-3 and OVCAR-8 (human ovarian cancer), HepG2 (human liver cancer), A2058, WM983b, HT168-M1/9 and M24 (human melanoma), B16 (mice melanoma), H1975, H1650 and A549 (human lung cancer), HT-29 (human colorectal adenocarcinoma), C26 (mice colon cancer) and PANC-1 (pancreatic cancer) were cultured in RPMI 1640 Medium with glutamine (Roswell Park Memorial Institute Medium, Lonza, Basel, Switzerland). Moreover, U87MG (human malignant glioma) and MRC-5 (normal human fibroblast) were cultured in Dulbecco’s Modified Eagle’s Medium (DMEM, Lonza) and PE/CA-PJ15 and PE/CA-PJ41 (head and neck cancer) were cultured in Iscove’s Modified Dulbecco’s Medium (IMEM, Sigma Aldrich, St. Louis, MO, USA). Apart from the cell culture medium for OVCAR-3 cells, all media were supplemented with 10% heat-inactivated FBS (Fetal Bovine Serum, Euroclone, Milan, Italy) and 1% Penicillin/Streptomycin (Sigma Aldrich), while OVCAR-3 cells were cultured in 20% FBS containing RPMI medium. The cell lines were mainly obtained from ATCC, except for PE/CA-PJ15, PE/CA-PJ41 (ECACC), A2058 (kindly provided by L.A. Liotta, NCI, Bethesda, MD, USA), HT-168-M1/9 (derived from A2058), M24 (kind gift from B.M. Mueller, Scripps Research Institute, La Jolla, CA, USA) and WM983b (kindly provided by Meenhard Herlyn, Wistar Institute, Philadelphia, PA, USA). Cells were cultured in sterile T25 or T75 flasks with ventilation cap (Sarstedt, Nümbrecht, Germany) at 37 °C in a humidified atmosphere with 5% CO_2_.

### 4.3. In vitro Anti-Proliferative Activity of GnRH-III-Dau Conjugates and Free Dau

For the evaluation of the in vitro anti-proliferative activity of the GnRH-III conjugates and free Dau, the cell viability was determined by MTT assay (3-(4,5-dimethylthiazol-2-yl)-2,5-diphenyl-tetrazolium bromide) that was obtained from Sigma Aldrich. After standard harvesting of the cells by trypsin-EDTA (Lonza), 4 × 10^3^ till 10 × 10^3^ cells per well depending on cell line, were seeded in serum containing growth medium to 96-well plates (Sarstedt) and incubated. After 24 h, the growth medium was removed and cells were treated with various concentrations of conjugates **1** and **2** (32 nM–100 µM) or free Dau (0.1 nM–10 µM), dissolved in serum free medium and incubated for 24 h under standard conditions. The control wells were treated with serum free medium. After 24 h of treatment, the cells were washed twice with serum free medium and then cultured in serum containing medium for an additional 48 h. Afterward, the MTT assay was performed, in order to determine cell viability, by adding 20 µL of MTT solution (5 mg/mL in PBS) to each well and after 2 h of incubation at 37 °C, the supernatant was removed. The formazan crystals were dissolved in 100 µL of a 1:1 solution of DMSO (Sigma Aldrich):EtOH (Molar Chemicals Kft., Halásztelek, Hungary) and the absorbance was measured after 15 min. at λ = 570 nm by using a microplate reader (Bio-Rad, model 550, Hercules, CA, USA). The IC_50_ values of the conjugates and free drug were calculated while using GraphPad Prism 6 (GraphPad Software, San Diego, CA, USA). The experiments were done in triplicate and each experiment was repeated twice.

### 4.4. RT-qPCR for GnRH-R mRNA Level of Expression

The GnRH-R mRNA expression of distinct cell lines was quantified via RT-qPCR. The total RNA was isolated while using Trizol^®^ reagent (Ambion, by Life Technologies, Carlsbad, CA, USA), followed by chloroform (Carlo Erba Reagents S.A.S., Vel-de-Reuil, France) extraction and isopropanol (Carlo Erba Reagents S.A.S.) precipitation. The purity and concentration of RNA were determined by measuring the absorbance at 260 nm and 280 nm (NanoDrop ND-1000, Wilmington, DE, USA). For cDNA synthesis a GeneAmp PCR System 9700 thermo cycler (Applied Biosystems, Waltham, MA, USA) was used. 250 ng total RNA from cell lines and human colon RNA sample (Ambion) were reverse transcribed while using Moloney murine leukemia virus reverse transcriptase (MMLV), oligo (dT)18 primer (Takara, Mountain View, CA, USA), reaction mix, dNTPs mix and recombinant RNase inhibitor (Clontech, San Francisco, CA, USA). After reverse transcription, the cDNA samples were stored at −20 °C until further processing. Human total cDNA (Clontech) was used as control. qPCR was run on the StepOnePlus^TM^ real-time PCR system in order to amplify the cDNA (Applied Biosystems). SYBR green primer assays were obtained for: Human GnRH-R gene with reference sequence (RefSeq) NM_000406(2) and exon location 1-1 (Integrated DNA Technologies (IDT), Skokie, IL, USA); human β-actin gene with RefSeq NM_001101 (Qiagen, Hilden, Germany); and, human GAPDH gene with RefSeq NM_002046(1) (Qiagen, Hilden, Germany). The average of the target GnRH-R gene was normalized to β-actin and GAPDH as endogenous housekeeping genes. As reference sample (presented as RQ value = 1 on the graph), MRC-5 cells (normal human lung fibroblasts) was chosen. Data are representative of two independent experiments.

### 4.5. Quantitative Western Blot Studies

The cells were lysed in lysis buffer (cOmplete™ Lysis-M; Roche, Mannheim, Germany). The protein concentration was determined by Pierce^TM^ BCA protein assay kit (Thermo Fisher Scientific, Rockford, IL, USA). Samples were denatured in Novex^TM^ LDS sample buffer and NuPage^TM^ sample reducing agent (Invitrogen, Carlsbad, CA, USA) at 70 °C for 2 min. in Eppendorf^®^ Thermo-mixer Compact (Hamburg, Germany). 30 µL protein sample (30 µg) were loaded to a 1 mm thick 4–12% Bis-Tris plus gel (Invitrogen^TM^) and run with Novex^TM^ NuPage^TM^ MES SDS buffer (Invitrogen^TM^) while using a Bio-Rad 1000/500 Constant Voltage Power Supply. Blotting was performed by iBlot Gel Transfer Stacks Kit (Invitrogen^TM^), whereby proteins were transferred to a nitrocellulose membrane while using a Invitrogen iBlot^TM^ dry blotting system. The membrane was incubated overnight at 4 °C with GnRH-R antibody (Proteintech, Rosemant, IL, USA, rabbit Polyclonal 19950-1-AP; 1:500 dilution). The secondary antibody (Cell Signaling Technology, Danvers, MA, USA, goat Anti-Rabbit-horseradish peroxidase (HRP) conjugated, 7074; 1:1000) was incubated 1 h at room temperature (r.t.). Amersham^TM^ ECL^TM^ prime western blotting detection reagent (GE Healthcare, Buckingamshire, UK) was used to visualize bands on a Bio-Rad ChemiDoc^TM^ MP Imaging System. The software Image Lab (Bio-Rad) was used to evaluate the signal level of the bands, followed by normalization to the β-actin signal (Cell Signaling Technology, rabbit mAb; HRP conjugated, D6A8; 1:1000). Normalized values were compared to value that was obtained from MRC-5 cells which arbitrary set as 1. Data are representative of two independent experiments.

### 4.6. GnRH-R Cell Surface Expression Level Determination by Flow Cytometry

Cells were harvested and one million cells from each cell line were used in the experiments. The cells were fixed with 4% Paraformaldehyde (PFA) for 10 min. at r.t., washed with phosphate buffered saline (PBS) and exposed to 3% bovine serum albumin (BSA) in PBS for 20 min. at r.t. Afterwards, GnRH-R antibody (Proteintech, rabbit polyclonal, 19950-1-AP) was used in a concentration of 0.2 µg/million cells (1:135 dilution), diluted in PBS and 3% BSA solution and incubated for 2 h at r.t. A fluorescent secondary antibody was used for detection (Cell Signaling Technology, Alexa 488-conjugated anti-rabbit IgG Fab fragment, CST 4412, 1:1000) and incubated at r.t. for 30 min. As control, samples only exposed to secondary antibody were used. The fluorescence was detected using the FITC-A channel of FACSVerse^TM^ Flow Cytometer (BD Biosciences, Franklin Lakes, NJ, USA). The BD FACSuit^TM^ software was applied to evaluate geomean fluorescence intensity. For each cell line, the ratio between GnRH-R geo mean fluorescence intensity (MFI) and secondary antibody control geo MFI was calculated. The ratio values from all cell lines were normalized to the results obtained from MRC-5 cells. The results of two experiments are presented.

### 4.7. Cellular Uptake of GnRH-III-Dau Conjugates by Flow Cytometry

The cells were seeded to 24 well plates in a cell density of 1.5 × 10^5^ cells per well, incubated 24 h at 37 °C and then treated with both conjugates (40 µM) for 6 h. Control wells remained untreated. After harvesting, cells were washed in PBS and quenched with 0.04% Trypan-Blue (Invitrogen). The fluorescence intensity was detected while using the PE-A channel of FACSVerse^TM^ Flow Cytometer (BD Biosciences) and geo MFI was evaluated using BD FACSuite^TM^ software. The geo MFI ratio between control and samples treated with conjugates was calculated in order to compare the uptake on different cell lines. Data are representative of two independent experiments.

### 4.8. Stability of GnRH-III-Dau Conjugates in Mice Plasma

The GnRH-III conjugates **1** and **2** were incubated in 90% mice plasma derived from healthy mice. Initially, the conjugates were dissolved in ddH_2_O and diluted with mice plasma (90%) to a final concentration of 10 µM. The mixture was incubated at 37 °C and aliquots were taken after 0, 0.5, 1, 2, 4, 8 and 24 h. The reaction was quenched by adding 10 µL acetic acid. High molecular weight plasma proteins were removed by ultra-centrifugal filters with a cut-off of 10 kDa (Merck Millipore, Darmstadt, Germany). The lower molecular weight fractions were analyzed by LC-MS. As control, 90% plasma plus 10% water, as well as 10 µM of bioconjugate in 100% water was incubated and analyzed in the same manner.

### 4.9. Experimental Animals

Adult female inbred BALB/c mice from a specified pathogen free (SPF) breeding of the National Institute of Oncology (Budapest, Hungary), were used in acute and chronic toxicity studies and in orthotopic 4T1 mice breast tumor model experiment. The mice were kept in a sterile environment in Makrolon^®^ cages at 22–24 °C (40–50% humidity), with a lighting regulation of 12/12 h light/dark. The animals had free access to tap water and were fed with a sterilized standard diet (VRF1, autoclavable, Akronom Kft., Budapest, Hungary) ad libitum.

The immunodeficient SCID mice on a C.B.-17 background were bred in specific opportunistic and pathogen free isolator breeding rooms. The breeding isolator was supplied with corn-cob bedding and standard VRF1 rodent chow and with acidified (pH 3) sterilized distilled water. The mice from the breeding rooms were used for the orthotopic model of human breast and human colon cancer. They were held in filter-top boxes in the experimental barrier rooms and every box-opening was performed under a Class 100 laminar-flow hood by an operator that was dressed in sterilized surgical attire. The cage components, corn-cob bedding and food (VRF1 from Special Diet Services) were steam-sterilized in an autoclave (121 °C, 20 min.). The animals used in these studies were cared according to the “Guiding Principles for the Care and Use of Animals” based upon the Helsinki declaration and they were approved by the ethical committee of National Institute of Oncology. Animal housing density was according the regulations and recommendations from directive 2010/63/EU of the European Parliament and of the Council of the European Union on the protection of animals used for scientific purposes [79]. Permission license for breeding and performing experiments with laboratory animals: PEI/001/1738-3/2015 and PEI/001/2574-6/2015.

### 4.10. Acute and Chronic Toxicity Studies of GnRH-III-Dau Conjugates

In order to determine the toxicity of the conjugates, in vivo acute and chronic toxicity studies were performed on healthy animals. In acute toxicity study, adult BALB/c female mice (20–23 g) were treated by a single intraperitoneal (*i.p.*) injection of conjugate **2**, whereby different doses (3.125, 6.25, 12.5, 25 and 50 mg/kg Dau content) were administrated to each groups (three mice per group). In chronic toxicity studies, adult female BALB/c mice (23–26 g) were treated either with GnRH-III conjugates by *i.p.* administration with a dose of 15 mg/kg Dau content on day 1, 3, 6, 8 and 10, or with free Dau (1 mg/kg) on day 1 and 8. In case of the control group, sterile water was administered. Each group consisted of three mice. The toxicity was evaluated on the basis of life span, behavior and appearance of the mice, as well as the body weight. Parameters were followed for 14 days.

### 4.11. Mouse Model of Orthotopic Mice Breast Carcinoma, Doses of Treatments and Measurements

Adult BALB/c female mice (20–25 g) were used in this experiment and kept under the same conditions, as described above. 4T1 mice breast carcinoma cells, maintained as described above, were orthotopically injected into the lower quarter of the right mammary fat pad line of mice, whereby 1 × 10^6^ cells were used per animal, suspended in 50 µL M199 medium (Sigma Aldrich). The treatment started seven days after cells inoculation, when the average tumor volume was 38 mm^3^, by *i.p.* administration of the conjugates or free Dau. Four groups with seven animals per group were established and treated with certain doses and schedules. The mice in the control group were treated with sterile water, while mice in free Dau group were treated with a dose of 1 mg/kg, once per week, on days 7, 14 and 21 after cells inoculation. The groups of conjugates **1** and **2** were treated on day 7, 10, 14, 17, 21 and 24 with a dose of 10 mg/kg Dau content. Animal weight and tumor volumes were measured initially when the treatment started and at periodic intervals according to the treatment schedule. A digital caliper was used to measure the longest (a) and the shortest diameter (b) of a given tumor. The tumor volume was calculated while using the formula V = ab^2^ × π/6, whereby a and b represent the measured parameters (length and width). The termination of the experiment was initiated 28 days after cell inoculation, respectively 22 days after treatment start, since the average volume of the tumors in the control group reached 1800 mm^3^. The mice from all groups were sacrificed by cervical dislocation. The anti-tumor effects and the liver toxicity of the conjugates and free Dau were evaluated based on the tumor volume and the liver weight/body weight ratio in each group. Moreover, the anti-proliferative and anti-metastatic activity of the conjugates and free Dau was evaluated in primary tumor and in metastases on the peripheral organs.

### 4.12. Mouse Model of Orthotopic Human Breast Carcinoma, Doses of Treatments and Measurements

Adult SCID female mice (22–28 g) were used in this experiment and kept under the same conditions, as described above. MDA-MB-231 human breast carcinoma cells were injected orthotopically into the lower quarter of the right mammary fat pad line of mice, whereby 1.5 × 10^6^ cells were used per animal, suspended in 50 µL of M199 medium. The treatments started 21 days after cells inoculation, when average tumor volume was 40 mm^3^. Free Dau and conjugates were administrated by *i.p.* injection. Four groups with seven animals per group were established and treated with certain doses and schedules. The mice in the control group were treated with sterile water, while the mice in the free Dau group were treated with a dose of 1 mg/kg, once per week, on day 21, 28, 35 and 42 after cell inoculation. The groups treated with the conjugates **1** and **2** were treated on day 21, 24, 28, 31, 35 and 38 with a dose of 15 mg/kg Dau content. The last treatment was performed on day 42 after cell inoculation, whereby a dose of 7.5 mg/kg Dau content was applied. The animal weights and tumor volumes were measured when the treatment was initiated and at periodic intervals according to the treatment schedule. The tumor volume was calculated in the same manner as described for mice breast carcinoma. The experiment was terminated 45 days after cell inoculation (25 days after treatment initiation), due to bad conditions of animals in the control group. The mice from all groups were sacrificed by cervical dislocation and primary tumors and livers were harvested and weighed, while number of animals with metastases were counted. The anti-tumor effects of the conjugates were evaluated measuring the tumor volume and the tumor weights, while the toxicity effects of conjugates were evaluated measuring liver weights and calculating the liver weight/body weight ratio.

### 4.13. Mouse Model of Orthotopic Human Colon Cancer, Doses of Treatments and Measurements

#### 4.13.1. Development of the Primary Tumor for Transplantation

Immunodeficient, 6–8 weeks old SCID female mice (21–27 g) were used in this experiment. HT-29 colon carcinoma cells were subcutaneously injected into one side of the intrascapular region, whereby 2 × 10^6^ cells were used per animal, suspended in 200 µL of M199 medium, in order to establish the xenografts with primary tumor. After two weeks, the mice with palpable tumors were sacrificed by cervical dislocation, disinfected with iodine and the subcutaneous tumor was aseptically dissected out. Tumor pieces of 2 mm^3^ were transplanted orthotopically under aseptic conditions into anesthetized (narcotic mixture: tiletamine 14.8 mg/kg, zolazepam 14.8 mg/kg, xylazine 10 mg/kg and butorphanol 2.4 mg/kg) SCID female mice.

#### 4.13.2. Tumor Transplantation

The abdomen of mice was disinfected with iodine and alcohol, a small midline incision (0.5 cm) was then made and the colorectal part of the intestine was exteriorized. Serosa of the site where the tumor pieces should be implanted were removed. Tumor tissue fragments of HT-29 human colon tumor were implanted on the top of the animal intestine, whereby an 8/0 surgical (polypropylene) suture was used to suture it on the wall of the intestine. The intestine was returned to the abdominal cavity and the abdominal wall was closed with 4/0 surgical (polyglycolic acid) sutures. The wound was disinfected with iodine and alcohol again and the animals were kept in a sterile environment. On the next day, no sign of pain and/or stress of the mice could be observed.

#### 4.13.3. Doses and Treatments

The treatments started seven days after tumor transplantation by *i.p.* administration of the free Dau and conjugates which were dissolved in sterile water for injection. Six mice per group were used. One group of mice was treated with free Dau at a dose of 1 mg/kg body weight on day 7, 13 and 20 after tumor transplantation. The groups of **1** and **2** were treated with a dose of 10 mg/kg Dau content on day 7, 10, 13, 16, 20, 23 and 27 after tumor transplantation. The control group was treated with sterile water. The experiment was terminated on day 30 after tumor transplantation (day 24 of treatment). The Dau group was terminated already on day 23 after tumor transplantation (day 17 of treatment), since the animals revealed a significant loss of weight. The mice from all groups were sacrificed by cervical dislocation. Their tumors and livers were harvested and weighed.

### 4.14. Immunohistochemical Staining of KI-67

The routinely formalin-fixed tumors were dehydrated in a graded series of ethanol, infiltrated with xylene and embedded into paraffin at a temperature not exceeding 60 °C. Two micron thick sections were mounted on Superfrost slides (Thermo Shandon, Runcorn, UK) and then manually deparaffinized. To block endogenous peroxidase activity, slides were treated for 20 min. at r.t. with 3% H_2_O_2_ in methanol. The slides were immersed in 6% citrate buffer (pH = 6) and exposed to 98 °C water bath for 40 min. Afterwards, the slides were primarily treated with antibody against human KI-67 (Dako, Glostrup, Denmark, 1:40) and then incubated for 1 h at r.t. After washing, secondary antibody Biotinylated Link (Dako) was applied for 10 min. at r.t. For visualization, supersensitive one step polymer HRP (Biogenex, Fremont, CA, USA) was used with 3-amino-9-ethylcarbazole (AEC) as chromogen. Staining without the primary antibody served as the negative control.

### 4.15. Scoring of Proliferation Index, Micro and Macro-Metastases

Liver, spleen, lung and kidneys were harvested and fixed in formalin. All of the visible macro-metastatic lesions of the peripheral organs of seven animals from the control and treated groups on a stereo microscope Kruss MSZ5600 (Kruss Optronic, Hamburg, Germany) under 7–45-fold magnification were counted. The percentages of macro-metastatic lesions in the treated groups compared to the control group was calculated.

Proliferation marker stained samples were evaluated on light microscope Olympus BH-2 microscope (Olympus, Tokyo, Japan). The proliferation index in primary tumors and in lung metastases were determined counting the KI-67-positive tumor cells manually per field of view under light microscope (400-fold magnification), whereby five fields of vision per sample from three animals were evaluated. The proliferation index was calculated as percentage of KI-67 positive cells from all cells in the field of view.

The number of micro-metastases, which were KI-67 positive stained in the lung samples sections on microscopic slides were manually counted per field of view under light microscope (100-fold magnification), whereby five fields of vision per sample from three animals were evaluated.

### 4.16. Statistical Analysis

The statistical analyses were performed by GraphPad Prism 6 (GraphPad Software) using the non-parametric Mann-Whitney (independent samples) test. Statistical analysis for uptake studies of conjugates was performed by two-way ANOVA test. The experimental data were filtered by Gaussian statistics, where *p*-values lower than 0.05 were considered statistically significant.

## 5. Conclusions

We could demonstrate that both GnRH-III-Dau conjugates possess efficient growth inhibitory effect on various cancer cells, whereby the biological activity is strongly connected to the expression of GnRH-Rs. Different in vitro studies pointed out that cells with lower receptor expression level remain less affected by the conjugates than cells with higher GnRH-R expression level. This ensures the selectivity of the compounds to GnRH-R positive cancer cells, which is of high relevance for the therapeutic success of targeted chemotherapy. Encouraged by the promising in vitro results, we studied the in vivo anti-tumor activity of compounds **1** and **2** on tumor bearing mice. Therefore, three different orthotopic mice models have been established including 4T1 mice and MDA-MB-231 human breast carcinoma, as well as HT-29 human colorectal cancer. In general, we could clearly show that the treatment with the GnRH-III-Dau conjugates elicit a significant in vivo tumor growth inhibitory effect in all three mice models, while toxic side-effects were substantially reduced in comparison to the treatment with free Dau. This indicates clearly that the administration of our GnRH-III based DDSs provide valuable benefits over the application of the free drug. Moreover, the anti-metastatic effect of the conjugates on human and murine breast cancer bearing mice was significantly improved in comparison to the free drug, whereby especially the novel conjugate (Glp-d-Tic-Lys(Bu)-His-Asp-Trp-Lys(Dau=Aoa)-Pro-Gly-NH_2_; **2**) exhibited a reduced metastasis development in the spleen, lung and liver in the 4T1 murine BC model. Apart from that, compound **2** revealed a significant higher tumor growth inhibitory effect on orthotopically developed HT-29 human colon carcinoma bearing mice than the free Dau and **1**. All of these findings confirm that our novel lead compound **2** is a promising candidate for targeted tumor therapy in both colon cancer and metastatic breast cancer.

## Figures and Tables

**Figure 1 ijms-20-04763-f001:**
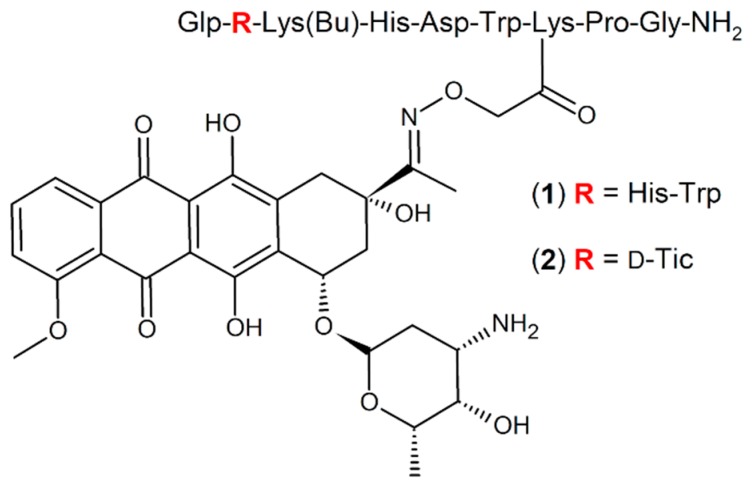
Structure of GnRH-III-Dau conjugates **1** and **2**.

**Figure 2 ijms-20-04763-f002:**
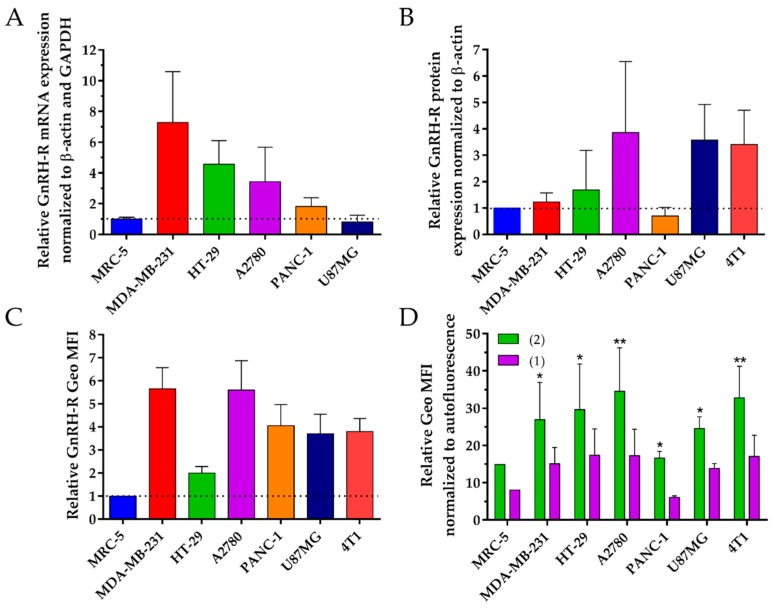
(**A**) GnRH-R mRNA expression level compared to MRC-5. (**B**) Absolute protein level of GnRH-R compared to MRC-5. (**C**) GnRH-R surface expression level compared to MRC-5. (**D**) Cellular uptake of conjugates **1** and **2** (40 µM), after 6 h treatment of the cell lines. Bar graphs represent average of two individual experiments (data are shown as average ± SD). The dotted line represents relative value of normal cell line as reference sample (presented as 1 on the graph). Statistical analysis for uptake studies of conjugates **1** and **2** was performed by two-way ANOVA test. * and ** mean significant at *p* < 0.05 and *p* < 0.01, respectively.

**Figure 3 ijms-20-04763-f003:**
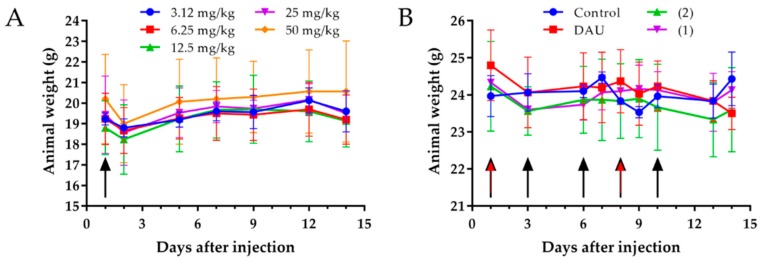
Animal body weight (grams, average ± SD). (**A**) Acute toxicity study of conjugate **2** with doses of 3.125, 6.25, 12.5, 25 and 50 mg/kg Dau content. (**B**) Chronic toxicity study of GnRH-III-Dau conjugates **1** and **2** with dose of 15 mg/kg Dau content, 5 treatments, black arrows; and free Dau 1 mg/kg, 2 treatments, red arrows. 3 mice per group.

**Figure 4 ijms-20-04763-f004:**
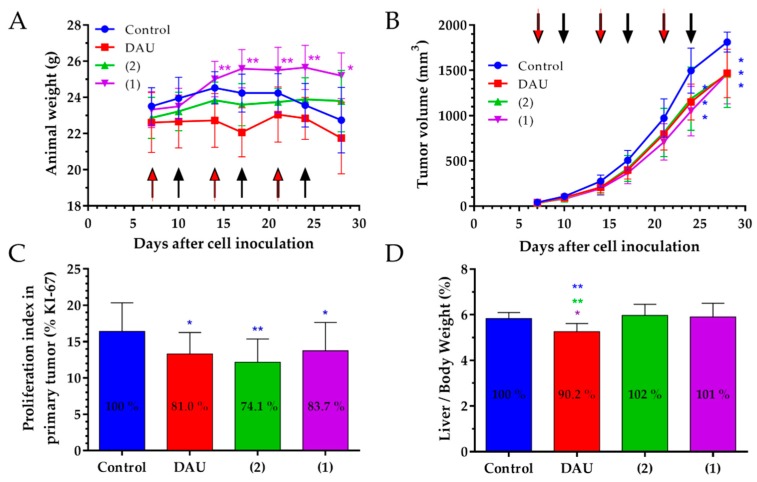
Effect of GnRH-III-Dau conjugates **1** and **2** (10 mg/kg Dau content, 6 treatments, black arrows) and free Dau (1 mg/kg, 3 treatments, red arrows) in orthotopic 4T1 mice breast carcinoma bearing mice. (**A**) Animal body weight (grams, average ± SD). (**B**) Tumor volume (mm^3^, average ± SD). (**C**) Proliferation in primary tumor (average proliferation index ± SD). (**D**) Liver weight/body weight ratio (percentage, average ± SD) after termination of experiment, 28 days subsequent to cells inoculation, 7 animals per group, statistical analysis was performed by Mann–Whitney test. * and ** mean significant at *p* < 0.05, *p* < 0.01 and *p* < 0.001, respectively. * blue, purple, and green mean significant difference in control, **1**, and **2** groups, respectively, at the end of experiment compared to the start (**A**), and significant difference compared to control, **1**, and **2** groups respectively (**B**–**D**).

**Figure 5 ijms-20-04763-f005:**
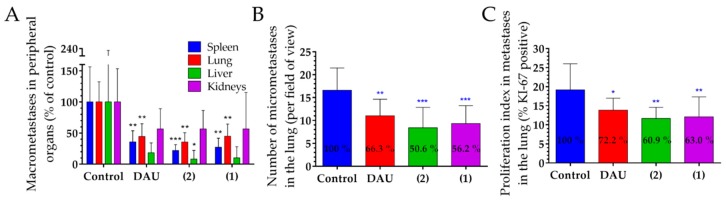
Effect of GnRH-III-Dau conjugates **1** and **2** (10 mg/kg Dau content, 6 treatments) and free Dau (1 mg/kg, 3 treatments) in orthotopic 4T1 mice breast carcinoma bearing mice. (**A**) Number of macro-metastases in peripheral organs (average % of control ± SD). (**B**) Effect of GnRH-III-Dau conjugates on the number of micro-metastases in lung (average number ± SD). (**C**) Proliferation of metastases in lung (average proliferation index ± SD). Statistical analysis was performed by Mann–Whitney test. *, ** and *** mean significant at *p* < 0.05, *p* < 0.01 and *p* < 0.001, respectively. * blue mean significant difference compared to control group (**B**,**C**).

**Figure 6 ijms-20-04763-f006:**
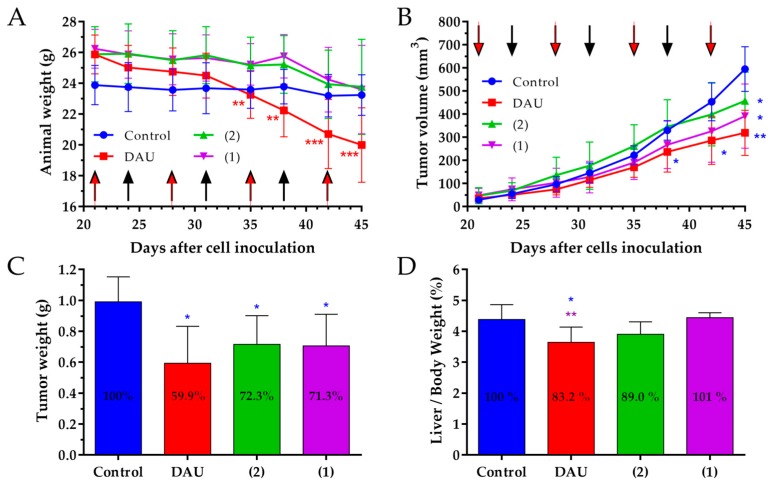
Effect of GnRH-III-Dau conjugates **1** and **2** (15 mg/kg Dau content, 6 treatments; 7.5 mg/kg Dau content of each conjugate, respectively, 1 treatment, black arrows) and free Dau (1 mg/kg, 4 treatments, red arrows) in orthotopic MDA-MB-231 human breast carcinoma bearing mice. (**A**) Animal body weight (grams, average ± SD). (**B**) Tumor volume (mm^3^, average ± SD). (**C**) Tumor weight (grams, average ± SD) after termination of the experiment, 45 days subsequent to cell inoculation. (**D**) Liver weight/body weight ratio (percentage, average ± SD) after termination of the experiment, 45 days after cell inoculation, 7 animals per group, statistical analysis was performed by Mann–Whitney test. *, ** and *** mean significant at *p* < 0.05, *p* < 0.01 and *p* < 0.001, respectively. * blue, red, and purple mean significant difference in control, Dau, and **1** groups, respectively, at the end of experiment compared to the start (**A**), and significant difference compared to control, Dau, and **1** groups respectively (**B**–**D**).

**Figure 7 ijms-20-04763-f007:**
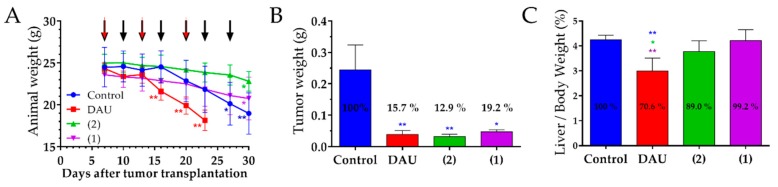
Effect of GnRH-III-Dau conjugates **1** and **2** (10 mg/kg Dau content, 7 treatments, black arrows) and free Dau (1 mg/kg, 3 treatments, red arrows) in orthotopic HT-29 human colon carcinoma bearing mice. (**A**) Animal body weight (grams, average ± SD). (**B**) Tumor weight (grams, average ± SEM) of mice from control, Dau, **1** and **2** groups after termination of experiment, 30 days after transplantation. (**C**) Liver weight/body weight ratio (percentage, average ± SD) of mice from control, Dau, **1** and **2** groups after termination of the experiment, 30 days after transplantation. Tumor and liver weight of mice from free Dau group on day 23 subsequent to transplantation, 6 animals per group, statistical analysis was performed by Mann–Whitney test. * and ** mean significant at *p* < 0.05 and *p* < 0.01 respectively. * blue, red, purple, and green mean significant difference in control, Dau, **1**, and **2** groups, respectively, at the end of experiment compared to the start (**A**), and significant difference compared to control, Dau, **1**, and **2** groups respectively (**B**,**C**).

**Table 1 ijms-20-04763-t001:** Anti-proliferative effect of free drug Dau and GnRH-III-Dau conjugates **1** and **2** on various cell lines.

Tumor Type	Cell Line	IC_50_ ^1^ 24 h + 48 h			Relative Potency ^2^	
		Dau (nM)	1 (µM)	2 (µM)	1/Dau	2/Dau
Breast	MDA-MB-231	54.6 ± 7.4	5.8 ± 0.8	1.9 ± 0.2	106.2	34.8
Breast	MCF-7	63.9 ± 21.0	16.5 ± 1.2	4.0 ± 0.8	258.2	62.6
Mice breast	4T1	56.0 ± 14.7	6.3 ± 0.9	1.8 ± 0.1	112.5	32.1
Colon	HT-29	202.9 ±1.0	15.5 ± 1.7	7.3 ± 0.3	76.4	36.0
Mice colon	C26	117.5 ± 8.6	10.6 ± 0.2	2.6 ± 0.7	90.2	22.1
Prostate	DU145	16.3 ± 4.6	5.3 ± 0.4	2.1 ± 0.2	325.2	128.8
Prostate	PC-3	32.7 ± 4.7	6.3 ± 0.3	2.4 ± 0.6	192.7	73.4
Glioblastoma	U87MG	126.4 ± 53.7	9.0 ± 0.8	2.3 ± 0.1	71.2	18.2
Ovarian	A2780	10.4 ± 1.6	1.4 ± 1.1	2.1 ± 0.5	134.6	201.9
Ovarian	OVCAR-3	404.0 ± 9.4	46.0 ± 1.3	8.2 ± 0.5	113.9	20.3
Ovarian	OVCAR-8	185.6 ± 99.8	5.7 ± 0.8	9.5 ± 0.8	30.7	51.2
Liver	HepG2	22.9 ± 1.4	6.8 ± 0.3	2.2 ± 0.7	296.9	96.1
Melanoma	A2058	35.1 ± 14.9	8.4 ± 0.3	2.6 ± 0.5	239.3	74.1
Melanoma	WM983b	49.8 ± 22.9	12.7 ± 1.5	2.6 ± 0.6	255.0	52.2
Melanoma	HT168-M1/9	27.5 ± 9.1	13.5 ± 1.1	2.9 ± 0.6	490.9	105.5
Melanoma	M24	118.8 ± 25.0	16.2 ± 0.2	3.5 ± 0.6	136.4	29.5
Mice melanoma	B16	26.0 ± 8.0	3.2 ± 0.8	1.1 ± 0.2	123.1	42.3
Head and neck	PE/CA-PJ41	45.6 ± 33.5	4.7 ± 0.8	1.7 ± 0.5	103.1	37.3
Head and neck	PE/CA-PJ15	50.5 ± 38.7	7.4 ± 0.8	2.9 ± 0.6	146.5	57.4
Lung	H1975	20.9 ± 2.7	4.1 ± 0.1	2.3 ± 0.7	196.2	110.0
Lung	H1650	50.3 ± 13.4	10.5 ± 1.1	4.0 ± 0.8	208.7	79.5
Lung	A549	69.3 ± 23.5	9.7 ± 0.6	4.3 ± 0.4	140.0	62.0
Pancreas	PANC-1	525.9 ± 24.7	>100	56.4 ± 4.5	>190.2	107.2
Normal fibroblast	MRC-5	287.6 ± 35.1	41.9 ± 3.8	19.7 ± 1.2	145.7	68.5

^1^ IC_50_ values (average ± SD). ^2^ Relative potency = IC_50_ conjugate / IC_50_ Dau.

**Table 2 ijms-20-04763-t002:** Anti-metastatic effect of free Dau and GnRH-III conjugates **1** and **2** in orthotopic MDA-MB-231 human breast carcinoma bearing mice on number of animals with metastases close to the primary tumor.

Treatment Groups of Mice	Number of Mice with Metastases
Control	5/5
Dau	4/7
1	3/7
2	3/7

**Table 3 ijms-20-04763-t003:** Summary of GnRH-R expression level and effect of free drug Dau and GnRH-III-Dau conjugates **1** and **2** on 4T1, MDA-MB-231 and HT-29 in in vitro, in vivo and ex vivo models.

Model Type	Experiment		4T1		MDA-MB-231			HT-29		
in vitro	mRNA GnRH-R					+ + +			+ +	
	Protein GnRH-R		+ + +			+			+ +	
	Cell surface GnRH-R		+ +			+ + +			+	
	Uptake	+ +	+ + +		+ +	+ + +		+ +	+ + +	
	Cytotoxicity (24+48h)	6.3 µM	1.8 µM	56 nM	5.8 µM	1.9 µM	55 nM	15.5 µM	7.3 µM	203 nM
	**Treatment**	**1**	**2**	**Dau**	**1**	**2**	**Dau**	**1**	**2**	**Dau**
in vivo	Acute toxicity	NT up to 50 mg/kg Dau content								
	Chronic toxicity	NT on 15 mg/kg Dau content; NT 1 mg/kg free Dau								
	Tumor inhibition	S* (19%)	S* (19%)	S* (19%)	S* (34%)	S* (23%)	S** (46%)	S* (81%)	S** (87%)	S** (84%)
	Animal weight	NT	NT	NT	NT	NT	T***	T* ^1^	T* ^1^	T*** ^1^
	Liver toxicity	NT	NT	T**	NT	NT	T*	NT	NT	T**
	Macrometastases in peripheral organs	S**	S***	S**						
	Number of mice with metastases close to the primary tumor				3/7 ^2^	3/7 ^2^	4/7 ^2^			
ex vivo	Proliferation index in primary tumor	S* (16%)	S** (26%)	S* (19%)						
	Number of micrometastases in the lung	S*** (44%)	S*** (49%)	S** (34%)						
	Proliferation index in metastases in the lung	S** (37%)	S** (39%)	S* (28%)						

^1^ Animal weight in control group was significantly reduced (T**). ^2^ All mice in control group had metastases close to the primary tumor. NT = non-toxic. T = toxic. S = significant inhibition. “+” represents level of expression or uptake. * represents level of significance. A higher level (double or triple + + +, ***). % represents percentage of inhibition compared to the control group.

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
