# Peer review of "Improved In Vivo Anti-Tumor and Anti-Metastatic Effect of GnRH-III-Daunorubicin Analogs on Colorectal and Breast Carcinoma Bearing Mice"

_ijms, 2019, doi:10.3390/ijms20194763_

Round 1

Reviewer 1 Report

The manuscript is very well written. The data are interesting and have been discussed with current literature. I recommend acceptance of the manuscript for publication.

Author Response

The manuscript is very well written. The data are interesting and have been discussed with current literature. I recommend acceptance of the manuscript for publication

Dear Reviewer 1,

Thank you very much for reviewing our manuscript. We are very pleased that Reviewer 1 likes our manuscript „Improved in vivo anti-tumor and anti-metastatic effect of GnRH-III-Daunorubicin analogs on colorectal and breast carcinoma bearing mice” and we greatly appreciate the referees’ nice comments. We are happy that Reviewer 1 recommended acceptance of our manuscript for publication in the journal International Journal of Molecular Sciences.

Reviewer 2 Report

General comments:

The aim of the study is to prove the antitumor activity (and anti-metastatic effect) of two conjugates combining GnRH-III peptide and Daunorubicin on three different in vivo models of breast and colon cancers.

The paper shows many results obtained in vitro on different cell lines and on mouse tumor models.

The paper is well written and well readable even if some information is missing to understand and to be convinced by the results particularly in terms of methods.

Specific comments:

1/Table 1: Please explain more the notion “relative potency” in the text and how to interpret the values. It is not clear for me.

2/Figure 2: First, in the legend, authors show the data as an average of two values. To my knowledge it is not correct to calculate an average with less than 3 values. Histograms could be changed into a table with both values for each cell line.

Figure 2 panel D, no statistics are provided to compare the uptake of cell lines and only at 6h of treatment. Did the authors a kinetic of capture for each cell line (30 minutes, 3h, 6h, 24h)?

Have the cells been treated by the same batches of conjugates? Or if it is not are we sure that the fluorescence of one conjugate is the same for all the cells treated?

3/ The authors show anti-tumor activity of both conjugates on three in vivo models but they did nor justified the rationale of each cancer model: for example, why choosing 4T1 and MDA-MB-231 orthotopic model? And for the HT29-based model? We supposed that it is a matter of aggressiveness or metastatic potential, but it is not clearly explained.

4/Figure 4: the proliferation index is provided using the Ki67 staining in tumors. The control tumors have a proliferation index of 16% that is not very high and the effect of drugs appears statistically significant but the proliferation index only decreased to 13%. It is not very high. What about the level of necrosis or fibrosis?

5/For all the experiments showing micro or macrometastases, please provide more explanation on the methods to detect and to count the lesions.

6/The tolerance (acute toxicity) of the treatment by the GnRH-III conjugates has been evaluated but only by the weight of the animals along the experiment and the weight of liver at the end of the treatment.

What about circulating levels of aminotransferase? Histology of liver?

What about the spleen weight?

Finally, as many models are used, in the discussion the authors could add a table to summarize all the results and to show the specificity of each model to highlight the interest of the conjugates.

Author Response

Dear Reviewer 2,

Please also see the attachment for point-by-point response.

Best regards.

Responses to the comments of Reviewer 2

Please note that the numbers of lines belong to changed manuscript (red line from left side of the manuscript in tracking changes mode). In square brackets are numbers of lines which you can see when option to see changes in the manuscript is “on” (grey line from left side).

1/Table 1: Please explain more the notion “relative potency” in the text and how to interpret the values. It is not clear for me.
Response: We agree with the referee. Therefore, we adjust the corresponding parts and added a more detailed explanation to the manuscript (see line 144-147 [144-147] and 348-349 [358-359]). The relative potency represents the ratio of conjugate’s IC50 and free Dau’s IC50 in order to show the potency of the conjugates as well as its targeting capacity independently from the cell line, due to different activity of free Dau that can enter cells non-specifically by passive diffusion. A lower value of relative potency indicates that the conjugate’s IC50 value is closer to the free Dau’s IC50 value, which implies that the targeting capacity of the conjugate as well as its anti-tumor effect is stronger on a particular cell line, compared to a cell line with higher relative potency. Higher values of relative potency indicate loss of potency of the conjugates with respect to free drug Dau (Rivas et al., European Journal of Organic Chemistry 2018, 2018 (23), 2902–2909; Kiss et al. Eur J Med Chem 2019, 176, 105–116.). Beneficial relative potencies (low values) of compounds 1 and 2 were obtained on breast and colon cancer cell lines and due to this they were chosen for further investigation. On the contrary, we obtained higher IC50 and relative potency values in case of MRC-5 and PANC-1 cells. Therefore we selected also these two cell lines for further investigations to gain a deeper insight into the correlation of the GnRH-R expression level and the biological activity of conjugates.

2/Figure 2: First, in the legend, authors show the data as an average of two values. To my knowledge it is not correct to calculate an average with less than 3 values. Histograms could be changed into a table with both values for each cell line.
Response: Thank you for the comment. We agree with the reviewer that the legend is not fully correct and we modified the term “replicates” into “individual experiments” (see line 175 [176]). The data presented in bar graph display the average of two values from two independent experiments. For GnRH-R mRNA expression level we had two parallels in each experiment, and according to this the average value was calculated based on 4 values (2 parallels from 2 individual experiments). In case of the two flow cytometry experiments, where we analyzed the cell surface GnRH-R expression level and the cellular uptake of the conjugates, each value represents the average of two individual experiments including 10000 events each.
It was our intention to present the data in an informative, direct and plain way, therefore we decided to show the level of GnRH-R expression and uptake of conjugates in a bar graph. From our point of view a bar graph provides an adequate and direct comparison of the difference between the cell lines. Nevertheless, in order to provide more comprehensive information we added a table with both values of the individual experiments to the supporting information (Table S1) (see line 825-
826 [840-841] in the manuscript). The corresponding modification can be found in lines 161 [161], 167 [167], 181 [182] and 188 [189] of the manuscript.

Figure 2 panel D, no statistics are provided to compare the uptake of cell lines and only at 6h of treatment. Did the authors a kinetic of capture for each cell line (30 minutes, 3h, 6h, 24h)?
Response: Thank you for the suggestion to show statistic of the uptake rates of both conjugates to provide a better comparison. To test the significance of the data, we performed a 2way ANOVA test (see modified Figure 2 panel D line 171 [172], and the text in line 175-177 [176-178], 395-398 [405-408] and 800-801 [815-816]). We could demonstrate that the cellular uptake of compound 2 was significantly higher than the uptake of conjugate 1 for each cancer cell line, but not for the normal cell line MRC-5, where we could not obtain a significant difference between the conjugates. This indicates that the uptake capacity on cancer cells is higher for conjugate 2 which is in line with our previous data (Schuster et al., Pharmaceutics. 2018 Nov 9;10(4)).
We agree with the reviewer that it might be beneficial to analyze the cellular uptake of the compounds for each cell line at different time points. However, in this study, it was our aim to prove if the cellular uptake rates are in line with the determined receptor levels of the cell lines. In order to ensure better comparability, we selected a treatment time of 6 hours and a compound concentration of 40 μM. As we know from previous studies (Schuster et al., Pharmaceutics. 2018 Nov 9;10(4); Schuster et al. Beilstein J Org Chem. 2018 Apr 4;14:756-771; Hegedüs et al. Eur J Med Chem. 2012 Oct;56:155-65.), under these conditions a high number of living cells are Dau positive (>90%) and the majority of the conjugated Dau is located on its site of action in the nucleus. This might be of high relevance for this study, since the fluorescence intensity can be also affected by intracellular processes (e.g. degradation state of the compound in lysosomes) and the subcellular localization of the conjugated Dau. Considering that the fluorescence intensity of the conjugated Dau can be influenced by the pH of the environment we intend to select a treatment time when the drug is predominantly located in a distinct cell compartment, in our case the nuclei, to ensure comparable pH values of the drug environment.

Have the cells been treated by the same batches of conjugates? Or if it is not are we sure that the fluorescence of one conjugate is the same for all the cells treated?
Response: Yes, the cells have been treated with the same batch of the conjugates and the same stock solution of the conjugates was used for all cell lines to provide a better comparability and to ensure a similar fluorescence intensity. We fully agree with the Reviewer that it is important for a direct comparison.

3/ The authors show anti-tumor activity of both conjugates on three in vivo models but they did not justified the rationale of each cancer model: for example, why choosing 4T1 and MDA-MB-231 orthotopic model? And for the HT29-based model? We supposed that it is a matter of aggressiveness or metastatic potential, but it is not clearly explained.
Response: We fully agree with the referee and included additional information about the selected in vivo models for breast cancer and why we chose them (see line 424-439 [434-450] for 4T1 and line 484-489 [496-501] for MDA-MB-231). Our motivation for the selection of the HT-29 model can be seen in line 520-532 [532-544], where we provide a comprehensive explanation. One reason was the obtained GnRH-R expression level of our study which is in line with data from literature. Moreover, we wanted to test conjugates on a cancer type which is not related to reproductive system. It was our aim to provide a tumor model which is not influenced by reproductive system related hormones.

4/Figure 4: the proliferation index is provided using the Ki67 staining in tumors. The control tumors have a proliferation index of 16% that is not very high and the effect of drugs appears statistically significant but the proliferation index only decreased to 13%. It is not very high. What about the level of necrosis or fibrosis?
Response: We understand the referee’s point of view and we fully agree that the control tumors have not a very high level of proliferation maker (KI-67) positive cells (16%), and that the drugs decreased the proliferation index only to 13%, which is not very high, but it is statistically significant. Both conjugates 1 and 2, as well as free Dau inhibited the tumor growth significantly by 19% in case of highly aggressive 4T1 mice breast tumor, and therefore the proliferation index was evaluated as parameter which supports this inhibition of tumor growth on a cellular level. Moreover, the inhibition of cancer cell proliferation elicited by the conjugates 1 and 2, as well as Dau was is same range as the obtained inhibition of tumor volume, 25, 16 and 18% respectively. By determining the proliferation index we wanted to strengthen our in vivo findings where we obtained that both of conjugates and free Dau decrease the proliferation capacity in primary tumor in same manner as we could detect for the inhibition of tumor growth. The level of necrosis or fibrosis was not determined.

5/For all the experiments showing micro or macrometastases, please provide more explanation on the methods to detect and to count the lesions.
Response: We totally agree with the reviewer that we should provide a more detailed explanation of the methods which we used to detect and to count the micro- and macro-metastatic lesions. We insert one more numbering paragraph to the materials and methods part of the manuscript, (4.15; Statistical analysis will be now 4.16), where we explained the methods how to detect and count the micro- and macro-metastases (see line 772 [784] and 783-798 [795-813]).

6/The tolerance (acute toxicity) of the treatment by the GnRH-III conjugates has been evaluated but only by the weight of the animals along the experiment and the weight of liver at the end of the treatment.
Response: We understand and endorse the Reviewers’ point of view that toxicity testing of new compounds is essential for drug development process and must be addressed to achieve the full potential of new drug delivery system. Therefore, we made a corrections in the manuscript (see line 413-415 [423-425] and 457-463 [468-474]). The preclinical toxicity testing on various biological systems reveals the species-, organ- and dose-specific toxic effects of an investigational product. The toxicity of substances can be observed by in vitro studies using cells/cell lines and in vivo exposure on experimental animals (Parasuraman, Journal of pharmacology & pharmacotherapeutics, 2011, 2(2), 74-9.). Animal weight and behavioral changes are the critical tool in toxicity testing as animals should be protected from stress and pain (Council NR. Guide for the Care and Use of Laboratory Animals. National Academies Press; 2010.). Dau is known to be rapidly and widely distributed in tissues, whereby the highest levels were found in the liver, spleen, kidneys, lungs and heart (Danesi et al., Eur J Cancer Clin Oncol. 1988, 24 (7), 1123–1131.). Since the liver is the vital organ in metabolism of Dau, production of a toxic intermediates which may trigger liver injury and impair with the liver function increasing the risk of toxicity (Paul et al., Cancer Lett. 1980, 9 (4), 263–269.). Analysis of organ weight in toxicology studies is an important factor for identification of potentially harmful effects of drugs (Michael et al., Toxicol Pathol. 2007, 35 (5), 742–750.), thus the liver weight analysis provides a better understanding of drug toxicity (Kuntzman et al., J. Pharmacol. Exp. Ther. 1966, 152 (1), 151–156.). To investigate toxic effects and the tolerance of the GnRH-III-Dau conjugates we determined the animal weight, behavioral changes and liver weight of the animals. We neither detected signs of behavioral changes nor significant body or liver weight changes administrating the GnRH-III-Dau conjugates, while significant changes have been obtained for the group treated with Dau. This may imply that the conjugates have a lower toxicity.

What about circulating levels of aminotransferase? Histology of liver?
Response: We understand the reviewer point of view that changes in the liver are often reflected by biochemical abnormalities of liver function. To analyze the level of circulating aminotransferases can serve as markers of hepatocellular injury (McGill, EXCLI J. 2016, 15:817–
828.) and provide useful information about the liver toxicity, but unfortunately we did not evaluate circulating levels of aminotransferase in the present study as well as the liver histology because we focused liver toxicity determination on liver weight as described above.

What about the spleen weight?
Response: We agree with the Reviewer that the spleen weight provides also a valuable information about the toxicity of the conjugates (see line 464-468 [475-479]). The spleen weight was measured in 4T1 tumor bearing mice. From the obtained data we can see that free Dau decreased the spleen weight significantly by 27.2% compared to the control group (Figure S2) (see line 827 [842] in the manuscript. This is in line with the obtained liver toxicity of free Dau measuring the liver weights. The treatment with conjugates 1 and 2 led also to a slightly decreased spleen weight, but in this case the decrease was not significantly indicating that the conjugates cause less harmful side effects then the free Dau.

7. Finally, as many models are used, in the discussion the authors could add a table to summarize all the results and to show the specificity of each model to highlight the interest of the conjugates.
Response: We appreciate that the Referee suggested to highlight the interest of the conjugates and we agreed to make a table in the manuscript where we show the specificity of each model (see line 321-323 [331-333], Table 3).

Reviewer 3 Report

The study by Randeloic et al. (Improved in Vivo Anti-Tumor and Anti-Metastatic Effect of GnRH-III-Daunorubicin Analogs on Colorectal and Breast Carcinoma Bearing Mice) evaluated the effects of novel GnRH-III-daunorubicin conjugate in experimental cancer in vitro and in vivo. The results showed both better uptake and inhibitory effects of this conjugate in comparison with free drug in various cancer cell lines depending on the GnRH-receptor expression status. Better efficacy and tolerability of conjugate was also confirmed in mice bearing breast carcinoma (4T1, MDA-MB-231) and colorectal carcinoma (HT-29 transplants). These initial results suggest that the GnRH-III is a promising candidate to be used as a drug carrier in cancer chemotherapy.

This is a comprehensive, reasonably designed study, with extensive data, the conclusions are clear and are adequately supported by results. There are several flaws, some of them may be attributed to the fact that this is a pilot study. I would like to make some comments and suggestions.

It can be assumed that the GnRH receptor positivity is necessary for the drug conjugate uptake and subsequent tumor growth inhibition, but still, the expression of GnRH-R should had been evaluated also in tumor samples and not only in vitro. For evaluation of tumor growth inhibition, only tumor weight/volume was considered in all three in vivo Proliferation and metastasizing was assessed only in mice bearing 4T1 carcinoma. Was there any particular reason why this was not assessed in other two experiments? In vivo experiments were terminated soon after the last dose of treatment, this is understandable in the case of free drug treatment which showed high toxicity. However, the conjugate therapy was well tolerated, thus it would be interesting to know how long did the effect of therapy last. Additional groups designed to monitor the survival would have made this clear. Liver toxicity was assessed only on the basis of liver weight. This is inadequate, in addition, the liver weight decrease might be attributed also to weight loss. Similarly, acute and chronic toxicity was evaluated on the basis of body weight too, and, besides, body weight gain would be more appropriate parameter. This should be considered in results and discussion. Line 824-828. The method of micro- and macro-metastases evaluation should be explained in more details, e.g. how many samples from each group were evaluated, how many fields in the case of macro-metastases were evaluated. A difference between micro- and macro-metastases should be clarified too. The number of animals per group was low. The authors used 7 animals for evaluation of treatment efficacy and only 3 animals for evaluation of toxicity, the latter number is absolutely insufficient for proper statistical analysis. Line 41: “Both cancers are highly malignant…” – this part should be rephrased. I believe this proclamation is too generalized, as breast cancer is a heterogenous disease and different types show different progression and prognosis. Line 294-296, Table 2. Both lines in the table provide the same information, either the first or the second line should be omitted. Line 725. The source for recommendation regarding the animal housing density should be cited.

Author Response

Dear Editor and dear Reviewer,

Thank you very much for reviewing our manuscript. Moreover, we greatly appreciate the referees’ comments and suggestions.

Responses to the comments of Reviewer 3

1. It can be assumed that the GnRH receptor positivity is necessary for the drug conjugate uptake and subsequent tumor growth inhibition, but still, the expression of GnRH-R should had been evaluated also in tumor samples and not only in vitro.

Response: The Referee’s question is reasonable assuming that the tumor inhibition effect of the conjugates is provided by uptake of the conjugates via GnRH receptors, and its high expression. Due to this we measured GnRH receptors level of expression and level of conjugates uptake in vitro (Figure 2 in the manuscript), in order to show correlation with in vivo tumor inhibition results for chosen tumor models. The presence of GnRH receptors and its expression in tumor samples and specimens are published in a lot of publications and due to this we did not evaluated it in our study:

GnRH-R expression in breast specimens: Miller et al., Nature. 1985, 313:231–3.; Eidne et al., Science. 1985; 229:989–991.; Fekete M, et al., J Clin Lab Anal. 1998, 3:137–47.; Nagy and Schally, Biol. Reprod. 2005, 73 (5), 851–859.; Emons et al., Trends in Endocrinology & Metabolism, 1997, 8(9): 355–362.; Reshkin et al., Int J Oncol. 1995, 7:371–5.; Moriya et al., Pathol Int. 2001, 51(5):333-7.; Kiesel et al., Geburtshilfe und Frauenheilkunde. 1988, 48: 420–424.; Kakar et al., Molecular and Cellular Endocrinology. 1994, 106(1-2): 145–149.; Baumann et al., Breast Cancer Res Treat. 1993, 25:37–46.

GnRH-R expression in triple negative breast cancer specimens: Buchholz et al., Int J Oncol. 2009, 35:789-796.

GnRH-R expression in 4T1 cells: Taheri et al., Int. J. Pharm. 2012, 431: 183–189.

GnRH-R expression in 4T1 tumor samples: Zoghi et al., Annals of Nuclear Medicine. 2016, 30(6):400–408.

GnHR-R expression in MDA-MB-231 cells: Kwok et al., Target Oncol. 2015, 10(3):365-73.; Seitz et al., BMC Cancer. 2014, 14:847.; Föst et al., Oncol Rep. 2011, 25(5):1481-7.; Lamharzi et al., Int J Oncol. 1998, 12(3):671-5.; Harris et al., Cancer Res. 1991, 51:2577–81.; Halmos G, et al., Cancer Letters. 1999, 136:129–36.; Eidne KA et al., J Clin Endocrinol Metab. 1987, 64:425–32.;

GnRH-R expression in MDA-MB-231 tumor samples: Kahán et al., Breast Cancer Res Treat. 2000, 59(3):255-62.

GnRH-R expression in colon specimens and HT-29 cells and tumor samples: Szepeshazi et al., Int J Oncol. 2007, 30(6):1485-92.

2. For evaluation of tumor growth inhibition, only tumor weight/volume was considered in all three in vivo. Proliferation and metastasizing was assessed only in mice bearing 4T1 carcinoma. Was there any particular reason why this was not assessed in other two experiments?

Response about proliferation: Due to the fact that mouse 4T1 breast carcinoma represents a clinically relevant triple negative breast cancer (Pulaski and Ostrand-Rosenberg, Curr Protoc Immunol. 2001, Chapter 20: Unit 20.2.), the same type as human MDA-MB-231 breast carcinoma, we evaluated proliferation capacity only in primary tumor of 4T1 carcinoma bearing mice in order to confirm significant inhibition of primary tumor volume due to the aggressive growth of 4T1 mice breast carcinoma (Tao et al., BMC Cancer. 2008, 8 (1): 228.) and on this way to support inhibition of tumor growth elicited with our GnRH-III-Dau conjugates.

Also we evaluated the proliferation capacity of GnRH-III-Dau conjugates and free Dau in primary tumor of HT-29 adenocarcinoma bearing mice where we obtained significant inhibition of proliferation index which confirms a significant inhibition of tumor growth (Figure 1. here bellow), but we did not present these results in the manuscript.

See the attached file for Figure 1

Response about metastases: It has been reported that in SCID mice, the remaining innate immune cells reduce the metastasis formation in distal organs [Dewan et al., Biomed. Pharmacother. 2005, 59 Suppl 2, S375-379; reference 76 in the manuscript]. Due to the absence of metastases in distal organs in orthotopic HT-29 and MDA-MB-231 tumors bearing mice we could not determine the inhibition effect of the compounds on metastases in distal organs. However, MDA-MB-231 tumor bearing mice developed metastases close to the primary tumor and we evaluated anti-metastatic capacity of the compounds by the number of animals which contain metastases close to the primary tumor in each group (Table 2. in the manuscript).

It is reported that 4T1 syngeneic breast cancer model in BALB/c mice is classified as metastatic on the basis of its ability to metastasize spontaneously and easily via the hematogenous route from the orthotopic site mainly to the lung, but also to liver and other distal organs and shares many characteristics with human breast carcinomas (Aslakson and Miller, Cancer Res. 1992, 52(6):1399-405.; Heppner et al., Breast Cancer Res. 2000, 2:331–334.; Bailey-Downs et al., PLoS One. 2014, 9(5):e98624.). Therefore, we were focused on the evaluation of the compounds’ anti-metastatic capacity on 4T1 tumor bearing mice as the best model with high metastatic and invasive properties, especially in the lung.

3. In vivo experiments were terminated soon after the last dose of treatment, this is understandable in the case of free drug treatment which showed high toxicity. However, the conjugate therapy was well tolerated, thus it would be interesting to know how long did the effect of therapy last. Additional groups designed to monitor the survival would have made this clear.

Response: We understand the referee’s approach and that it would be interesting to know how long did the effect of therapy last. As first, based on many regulations and recommendations following survival of animal until death is prohibited and experimental animals should be sacrificed by humane end-point suggestions. The Federation of European Laboratory Animal Science Associations (FELASA) recommends that the screening and developing of potential pharmaceutical agents and toxicity tests should be performed avoiding lethal endpoints (Baumans et al., Laboratory Animals.1992, 28(2), 97–112.).

Moreover, as stated in current regulatory testing guidelines document on “the recognition, assessment, and use of clinical signs as humane end-points for experimental animals used in safety evaluation” from the Organization for Economic Co-operation and Development (OECD), animals should be humanely killed rather than allowed to survive to the end of the scheduled study. The increased awareness of the ethical issues pertaining to animal experimentation requires that relevant scientific information and biological materials are obtained as early as possible, avoiding that animals have to reach severe stages of disease. (ENV/JM/MONO(2000)7)

According to regulations and recommendations from directive 2010/63/EU of the European Parliament and of the Council of the European Union on the protection of animals used for scientific purposes article 14 states that the methods selected for experimental procedures should avoid, as far as possible, death of the animals as an end-point due to the severe suffering experienced during the period before death, thereby allowing the animal to be sacrificed without any further suffering. Under Chapter III: Procedures, Article 13: Choice of methods. 3. Death as the end-point of a procedure shall be avoided as far as possible and replaced by early and humane end-points (Off J Eur Communities 2010, L276: 33–79.).

As second, by using the performed procedures we want to gather information at the same time to ensure a better comparison of treated groups and the control group. By sacrificing the control group before other groups we would not be able to compare the effect of conjugates on tumor and liver weight, and anti-proliferation and anti-metastatic capacity in a suitable way.

The mice in the 4T1 model were sacrificed since the average volume of the tumors in the control group reached 1800 mm3 as maximum tumor volume allowed by Guidelines for the welfare and use of animals in cancer research (Workman et al., British Journal of Cancer. 2010, 102:1555–1577.) (see line 725-727). Treated groups were terminated in order to obtain comparable data for effect of conjugates on liver weight, number of metastases in organs and proliferation index in same time point.

The mice in the MDA-MB-231 model were sacrificed, because free Dau caused a significant decrease of mice body weight by 20%. In addition two animals of the control group were in bad condition (see line 259-262). Although the mice in both conjugates treaded groups were in good condition, we decided to scarify them, thus we could compare the obtained data effect of conjugates on tumor and liver weight, as well as on metastases close to the primary tumor at same time point with the control group sacrificed by humane end-point recommendation.

The effect of conjugates on tumor growth in orthotopic HT-29 carcinoma bearing mice is only possible to follow by tumor weight after sacrificing of experimental animals. Due to this, sacrificing the control and the conjugates treated groups in same time point is necessary. In this experiment, the mice in free Dau treated group exhibited a significantly decreased body weight, thus the animals needed to be sacrificed by humane end-point. On the same day, the decrease of the body weights in the control and both conjugates treated groups were non-significantly lower. Therefore, we decided to continue experiment until the time point when the body weight of the mice in the control group was significantly decreased, when we sacrificed also and both conjugates group in order to compare effects of conjugates on tumor weight in same time point.

4. Liver toxicity was assessed only on the basis of liver weight. This is inadequate, in addition, the liver weight decrease might be attributed also to weight loss.

Response: We totally agree with the Referee that liver toxicity assessment based on liver weight is not totally adequate considering the fact that liver weight changing might be attributed also to body weight changing. According to this suggestion we calculated and evaluated liver weight / body weight ratio and made changes in the manuscript. (see line 220-224; 243-246; 269; 284-287; 309-313; 319; 465-466; 509-511; 523; 554-555; 560; 729; 751; Figure 4D; Figure 6D; Figure 7C; in Table 3. changed significance for liver toxicity in MDA-MB-231 and HT-29 model).

5. Similarly, acute and chronic toxicity was evaluated on the basis of body weight too, and, besides, body weight gain would be more appropriate parameter. This should be considered in results and discussion.

Response: In toxicity evaluation of compounds we have a panel of parameters that we follow during the procedure such as general look of the animal, quality of hair, mobility, facial expression, body condition, presence of pain and tremor, behavior and body weight (see line 191-198, 709-710). The body weight changes are the most informative (see line 416-417) because a decrease of the body weight by 20% compared to the start of experiment tell us about high toxicity of the tested compound. Due to not significant changes of body weight (in acute toxicity study animal weight stayed almost same as in the start of the experiment: body weight gain was up to 1.9%; in chronic toxicity study animal weight also stayed stable except for free drug DAU where decreasing of body weight was by 5%, but not significant), not significant increase, but not significant decrease, we explained and discussed our results according to this (see line 191-198, 415-426).

6. Line 824-828. The method of micro- and macro-metastases evaluation should be explained in more details, e.g. how many samples from each group were evaluated, how many fields in the case of macro-metastases were evaluated. A difference between micro- and macro-metastases should be clarified too.

Response: We totally agree with the reviewer that we should provide a more detailed explanation of the methods which we used to evaluate the micro- and macro-metastatic lesions.

The number of macro-metastases was evaluated counting all visible metastases under stereo microscope (under 7-fold up to 45-fold magnification) in spleen, lung, liver and kidneys in 7 animals from each group (see line 794-798).

Number of micro-metastases in lung, that were KI-67 positive stained, were counted on light microscope (under 100-fold magnification) in 3 animals from each group and 5 fields of vision per sample were evaluated (see line 805-807).

As macro-metastases we counted all lesions visible on stereo microscope, while micro-metastases we could only count on microscopic slides sections of the lung under light microscope.

7. The number of animals per group was low. The authors used 7 animals for evaluation of treatment efficacy and only 3 animals for evaluation of toxicity, the latter number is absolutely insufficient for proper statistical analysis.

Response: We understand the Reviewers’ point of view about the number of used animals in procedures, but in our experiments we follow recommendation that proper experimental design should reflect on methods that could reduce, refine and also replace (3Rs rule) the current techniques (Russel and Burch, Humane Society Press. 1959, 121–135.), where reduction approach implies that the number of animals employed in a given test should be minimized while still maintaining consistency and accuracy with scientific practices that would yield convincing and valid results (Robinson, School Sci Rev. 2005, 87:1–4.).

According to regulations and recommendations from directive 2010/63/EU of the European Parliament and of the Council of the European Union on the protection of animals used for scientific purposes article 11 states that the care and use of live animals for scientific purposes is governed by internationally established principles of replacement, reduction and refinement (3Rs rule). It is recommended that the numbers of animals used in experimental procedures may be reduced by implementing testing strategies that would reduce and refine the use of animals. By article 13 of the directive, the choice of methods have a direct impact on both the numbers of animals used and their welfare. Therefore experimental design should ensure the selection of the method that is able to provide the most satisfactory results and is likely to cause the minimum pain, suffering or distress of the animals. The methods selected should use the minimum number of animals that would provide reliable results and require the use of species with the lowest capacity to experience pain, suffering, distress or lasting harm that are optimal for extrapolation into target species (Off J Eur Communities 2010, L276: 33–79.).

By guidelines for accommodation and care of animals of European Convention for the Protection of Vertebrate Animals Used for Experimental and other Scientific Purposes (ETS No. 123; Council of Europe, 2006.) under Article 7 is stated that when a procedure has to be performed, a choice of procedures should be that selected one has to use the minimum number of animals, cause the least pain, suffering, distress or lasting harm and which are most likely to provide satisfactory results.

Therefore, we optimized the numbers of animals that were involved in our experiments. Based on today's extensive experience on acute toxicity testing of chemical substances, and on the outcome of specific studies from the past (Schütz and Fuchs, Arch.Toxicol. 1982, 51:197-220.; Lorke, Arch.Toxicol. 1983, 54:275-287.), it is now possible to conduct acute toxicity studies with the sacrifice of fewer animals (one to three per dose) and even to increase at the same time, the quality of the data obtained (Chinedu et al., Toxicol Int. 2013, 20(3): 224–226.; Erhirhie et al., Interdiscip Toxicol. 2018, 11(1): 5–12.). Same principle about reducing the number of animals we followed and in chronic toxicity study as in study of Aston et al. (BMC Cancer. 2017, 17(1):684.). Thus we used three animals per group in toxicity studies.

Treatment efficacy evaluation studies we performed also following the principle of reduction the number of animals as carried out in previously reported evaluation of GnRH compounds in MDA-MB-231 in vivo model (Föst et al., Oncol Rep. 2011, 25(5):1481-7; 5 mice per group); in HT-29 in vivo model (Szepeshazi et al., Int J Oncol. 2007, 30(6):1485-92.; 6-9 mice per group).; in 4T1 in vivo model (Calderon et al., Bioconjugate Chemistry. 2017, 28(2), 461–470.; 7 mice per group). 5 animals per group was used in Schubert et al. research (Breast Cancer Res Treat. 2011, 130:783–790.), and 6-10 animals per study group in treatment efficacy studies of Workman et al. (Br J Cancer. 2010 May 25; 102(11): 1555–1577.) Based on this and our previously published study of GnRH compounds treatment efficacy on HT-29 in vivo model (Kapuvári et al., Invest New Drugs. 2016, 34, 416–423.; 6-7 mice per group) we concluded that using 6-7 animals per group is adequate for investigation of treatment efficacy providing reliable results with reduced the number of animals.

8. Line 41: “Both cancers are highly malignant…” – this part should be rephrased. I believe this proclamation is too generalized, as breast cancer is a heterogenous disease and different types show different progression and prognosis.

Response: We agree with the Referee that breast cancer is a heterogenous disease with distinctive clinical, histopathological, biomarker and genetic heterogeneity where different types show different behavior, progression, prognosis and differently responses to therapy (Weigelt et al., Mol Oncol. 2010, 4:192–208.; Vargo-Gogola and Rosen, Nat Rev Cancer. 2007, 7(9): 659-72.; Simpson et al., J. Pathol. 2005, 205, 248–254.). According to this and Reviewers suggestion we rephrased the text in the manuscript (see line 41-43).

9. Line 294-296, Table 2. Both lines in the table provide the same information, either the first or the second line should be omitted.

Response: We totally agree with the Reviewer that both lines in the table 2. provide the same information, therefore we deleted “Metastases (%)” and left “Amount of mice with metastases” (see line 295, Table 2.).

10. Line 725. The source for recommendation regarding the animal housing density should be cited.

Response: We appreciate that the Referee suggested to cite the source for recommendation regarding the animal housing density. As previously shown that cage environments and animal density may significantly influence experimental outcomes (Hutchinson et al., ILAR Journal, 2005, 46(2), 148–161.), animal housing density in our experiments was according to regulations and recommendations from directive 2010/63/EU of the European Parliament and of the Council of the European Union on the protection of animals used for scientific purposes, Annex III about Requirements for establishments and for the care and accommodation of animals, Section B: Species-specific section, Table 1.1. on page 57 of the document (Off J Eur Communities 2010, L276: 33–79.). The manuscript was improved about it (see line 696-698) and the reference was introduced (see line 1066-1068).

Best regards,

Ivan

Round 2

Reviewer 3 Report

I approve of authors´corrections. I suggest that manuscript is checked to correct minor language issues.